# Two-step lookahead Bayesian optimization with inequality constraints

**Yunxiang Zhang**
Cornell University
yz2547@cornell.edu

**Xiangyu Zhang**
Cornell University
xz556@cornell.edu

**Peter I. Frazier**
Cornell University
pf98@cornell.edu

## Abstract

Recent advances in computationally efficient non-myopic Bayesian optimization offer improved query efficiency over traditional myopic methods like expected improvement, with only a modest increase in computational cost. These advances have been largely limited to unconstrained BO methods with only a few exceptions which require heavy computation. For instance, one existing multi-step lookahead constrained BO method [1] relies on computationally expensive unreliable brute-force derivative-free optimization of a Monte Carlo rollout acquisition function. Methods that use the reparameterization trick for more efficient derivative-based optimization of non-myopic acquisition functions in the unconstrained setting, like sample average approximation and infinitesimal perturbation analysis, do not extend: constraints introduce discontinuities in the sampled acquisition function surface. Moreover, we argue here that being non-myopic is even more important in constrained problems because fear of violating constraints pushes myopic methods away from sampling the boundary between feasible and infeasible regions, slowing the discovery of optimal solutions with tight constraints. In this paper, we propose a computationally efficient two-step lookahead constrained Bayesian optimization acquisition function (2-OPT-C) supporting both sequential and batch settings. To enable fast acquisition function optimization, we develop a novel likelihood-ratio-based unbiased estimator of the gradient of the two-step optimal acquisition function that does not use the reparameterization trick. In numerical experiments, 2-OPT-C typically improves query efficiency by 2x or more over previous methods, and in some cases by 10x or more.

## 1   Introduction

We consider constrained optimization of a continuous black-box function $f$ under continuous black-box constraints $g_i$, $\min_{x \in \mathbb{A}} f(x)$ subject to $g_i(x) \leq 0, i = 1, \ldots, I$, within the compact design space $\mathbb{A} \subseteq \mathbb{R}^d$. We suppose both $f(x)$ and $g_i(x)$ are derivative-free, time-consuming-to-evaluate and also noise-free. Such problems arise, for example, in tuning hyperparameters of machine learning models subject to runtime or fairness constraints and policy optimization in reinforcement learning with safety constraints. For instance, neural networks deployed on mobile phones must be accurate but may also have limited computation available while needing to respond to users in real time, creating a constraint on how long the model takes to predict at test time [2]. Another example, from [3], is predicting recidivism risk in the criminal justice system with a fairness constraint ensuring that false positive rates are equal across racial and ethnic groups. Other applications arise in drug discovery [4] and aircraft design [5].

Bayesian optimization (BO) has proven successful at solving black-box optimization problems with expensive objectives [6, 7], including constrained problems of the form above. BO methods for constrained problems include constrained expected improvement (EIC as in [8], rediscovered

35th Conference on Neural Information Processing Systems (NeurIPS 2021).

by [9]), constrained BO with stepwise uncertainty reduction [10], predictive entropy search with unknown constraints [11], Alternating Direction Method of Multipliers Bayesian optimization [12], constrained BO with max-value entropy search [13], and augmented Lagrangian techniques that convert constrained problems into a sequence of unconstrained ones [14]. It also includes methods designed specifically for equality and mixed constraints [15], for batch observations [16] and for problems with high dimensions [17].

All of these existing methods, however, for constrained Bayesian optimization (CBO), are myopic, in the sense that they only consider the immediate improvement in solution quality resulting from a function evaluation and ignore later improvements in solution quality enabled by this evaluation. (Notable exceptions are [1, 18], discussed below.) This greedy behavior may hinder an algorithm's ability to find good solutions efficiently. While a recent flurry of activity is addressing this issue for unconstrained problems [19, 20, 21, 22, 23], and the performance improvements provided by these non-myopic algorithms for unconstrained BO suggest that non-myopic BO is promising for constraints as well, substantial non-myopic development has not reached the constrained setting.

Moreover, we argue in detail below that being non-myopic provides even more value in constrained settings than it does in unconstrained ones. To find a global optimum under constraints quickly, an algorithm benefits by efficiently learning the boundary between the feasible region where the constraint is satisfied and the infeasible region where it is not. This is facilitated by sampling points likely to be close to this boundary. Myopic methods, however, such as EIC, undervalue sampling such points because they have a substantial probability of being infeasible and because infeasible points do not directly improve solution quality. Non-myopic methods, on the other hand, understand that learning about the boundary's location will provide future benefits, allowing them to value this information more appropriately. Localizing the boundary efficiently is especially important when the global optimum lies on this boundary, as it often does in constrained optimization when the objective (e.g., the quality of a product) is negatively correlated with a constraint (e.g., the cost required to produce it). We illustrate this via a simple example in §3 and our numerical experiments in §6, which show several-fold improvement over the state-of-the-art in some problems.

One existing non-myopic constrained BO method [1] first formulates CBO as a dynamic program (DP). However, this DP is intractable. To mitigate the issue, rollout, an approximate DP technique, is used. Nevertheless, this approach requires an extremely large amount of computation to approximate the multi-step lookahead policy well, especially in problems with more than a few dimensions, in part because it relies on computationally expensive derivative-free optimization , and because its acquisition function is computed via Monte Carlo, further increasing the computation required. This limits its applicability.

The constrained multi-information source BO method recently proposed by [18] is also non-myopic. It focuses on the multi-information source setting and assumes that the objective and constraint are evaluated in a decoupled fashion. In contrast, our applications of interest often compute the objective and constraints simultaneously. For example, when tuning ML hyperparameters to maximize accuracy subject to a model execution time constraint, the marginal cost of evaluating accuracy is negligible once we evaluate model execution time and incur the training cost. Thus, a method that evaluates the objective and constraints in a decoupled way discards information in such settings.

**Our Contributions.** We provide a novel non-myopic computationally efficient method for batch CBO. It substantially outperforms myopic CBO methods. In relation to [1] it requires substantially less computation to decide where to sample. Its query efficiency is substantially better in relatively higher-dimensional problems and is at least as good in lower-dimensional ones.

The key to our approach is a new method for optimizing stochastic acquisition functions, leveraging the likelihood ratio method [24]. Standard efficient approaches to optimizing stochastic non-myopic acquisition functions, such as infinitesimal perturbation analysis (IPA) [25] or the one-shot method [26] (also called sample average approximation or SAA), rely on a sampled acquisition function surface created using the reparameterization trick. In CBO, however, this surface is discontinuous, preventing the efficient use of these methods. Our novel approach is potentially generalizable to other settings where such discontinuities prevent the use of IPA and one-shot optimization.

Our work builds on the unconstrained two-step optimal method [21], overcoming substantial computational difficulties created by constraints' inherent discontinuities. These difficulties require abandoning the IPA approach used in [21] and instead developing a new likelihood-ratio-based

approach. The likelihood ratio method that we use here to estimate the gradient of the acquisition function relies on a change of measure of the same type used within importance sampling. [21] coincidentally also uses importance sampling, but in a fundamentally different way: as a variance reduction technique, and not for gradient estimation.

## 2    Background

We briefly review the literature on myopic CBO. Then we summarize standard results needed later on Gaussian processes and the widely used myopic method EIC as well as its batch version.

### 2.1    Myopic Constrained Bayesian Optimization

This work builds on the larger literature on myopic CBO. We review this literature here, giving more details than in §1. [9] proposes constrained expected improvement, a constraint-weighted expected improvement acquisition function which multiplies the expected improvement with the probability of feasibility associated with each constraint, rediscovering an approach due to [8]. [27] proposes an approach in which the point to sample is found by maximizing EIC and then a decision is made whether to evaluate the objective or the constraint based on the information gain. [10] proposes a stepwise uncertainty reduction method in which the acquisition function aims to maximally decrease our uncertainty on the location of the optimizer with a single evaluation. Predictive entropy search with constraints (PESC) [11], an extension of predictive entropy search [28] is another information-gain based approach. It chooses the point to evaluate by approximating the expected information gain on the value of the constrained minimizer. [14] proposes a hybrid approach combining the expected improvement with an augmented Lagrangian framework. [15] extends this technique by introducing an alternative slack variable formulation that handles equality and mixed constraints. Integrated expected conditional improvement (IECI) [29] proposes a new acquisition function that integrates a conditional improvement with respect to a reference point over the design space. [16] develops a quasi-Monte Carlo approximation of expected improvement under batch optimization with noisy observations and noisy constraints. [13] modifies the mutual information criterion of max-value entropy search [30] and extends to the constrained setting with the ability to handle both continuous and binary constraints. [12] leverages the ADMM framework to convert constrained problems into multiple unconstrained subproblems by introducing auxiliary variables for each constraint, then solves the subproblems using standard BO. Similar to TuRBO [31], [17] proposes an acquisition function for constrained optimization that scales to high dimensions by maintaining and adjusting trust regions.

### 2.2    Gaussian Processes

BO makes productive use of Gaussian processes (GPs) [32]. We put a GP prior on the objective function $f$, which is specified by a mean function $\mu(\cdot)$ and a kernel function $K(\cdot, \cdot)$. After observing the data points $D = \{x^{(1)}, x^{(2)}, x^{(3)}, \ldots, x^{(n)}\}$ and their corresponding function values $f(D) := \{f(x^{(1)}), f(x^{(2)}), f(x^{(3)}), \ldots, f(x^{(n)})\}$, the GP prior over $f$ is updated by:

$$f(x)|D, f(D) \sim \mathcal{N}\left(\mu(x; D), \sigma^2(x; D)\right),$$

where $\mu(x; D)$ is the posterior mean and $\sigma(x; D)$ is the posterior standard deviation. $\mu(D)$ is the set of values of the prior mean function evaluated at points in D.

While we support multiple constraints, we focus on a single constraint $g$ for ease of presentation. We use $\mu^c(\cdot)$ and $K^c(\cdot, \cdot)$ to denote the mean function and the kernel of the independent GP prior on $g$ respectively, and $\mu^c(x; D)$ and $\sigma^c(x; D)$ to denote the mean and standard deviation of the posterior. We refer readers to the supplement for multiple constraints.

### 2.3    Constrained Expected Improvement

We briefly review constrained expected improvement (EIC) [9, 8], introducing notation used later. Suppose we have independent GP priors on $f$ and $g$ and observations $f(D) := \{f(x) : x \in D\}$ and $g(D) := \{g(x) : x \in D\}$ at a collection of datapoints $D$. Let $f^*$ be the best point observed so far

subject to our constraints, $f^* = \min_{x \in D, g(x) \le 0} f(x)$. The constrained expected improvement at $x$ is:

$$\text{EIC}(x) = \mathbb{E}\left[ [f^* - f(x)]^+ \cdot \mathbb{1}\{g(x) \le 0\} \right] = \mathbb{E}[f^* - f(x)]^+ \cdot \mathbb{E}[\mathbb{1}\{g(x) \le 0\}] = \text{EI}(x) \cdot \text{PF}(x), \quad (1)$$

where $\mathbb{E}[\cdot]$ is the expectation taken with respect to the posterior given $D$, $f(D)$, and $g(D)$. $\text{EI}(x)$ and $\text{PF}(x)$ are the expected improvement and the probability of feasibility at $x$ respectively, and both have analytic forms. Here, we assume the prior on $f$ and $g$ are independent as in [9]. We refer readers to [33] and [9] for more details.

We now define EIC in the batch setting, building on the first discussion of which we are aware [16]. Let $\mathbf{X} = \{x^{(n+1)}, x^{(n+2)}, \ldots, x^{(n+q)}\}$ be the batch of $q$ candidate points that we consider evaluating next. Then the *constrained multi-points expected improvement* $\text{EIC}(\mathbf{X})$ is,

$$\text{EIC}(\mathbf{X}) = \mathbb{E}\left[ \max_{x \in \mathbf{X}} \ (f^* - f(x))^+ \cdot \mathbb{1}\{g(x) \le 0\} \right]. \quad (2)$$

The details of the derivation of $\text{EIC}(\mathbf{X})$ are provided in the supplement.

## 3 Constrained Two-step Acquisition Function

In this section, we first show that why being non-myopic is important in CBO. Then we formally define the novel acquisition function 2-OPT-C that is at the heart of our method.

### 3.1 The Importance of Being Non-Myopic in CBO

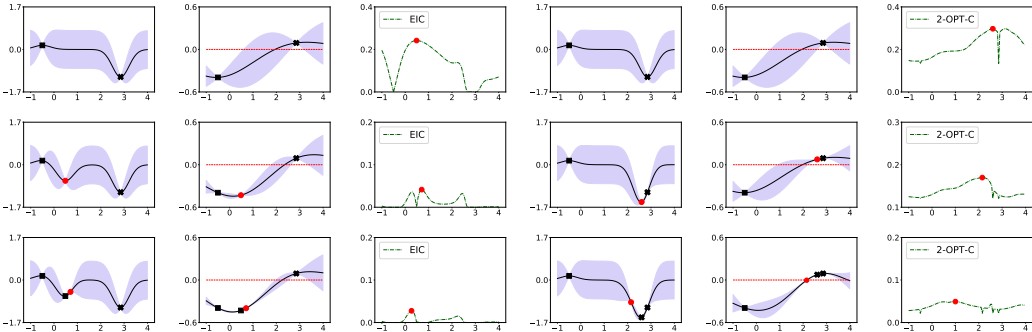

Figure 1: EIC and 2-OPT-C optimizing a 1-d synthetic function. The first three graphs of each row are the posterior on $f$, the posterior on $g$, and the acquisition function under EIC, and the last three graphs are the corresponding quantities under 2-OPT-C. Each row shows one iteration, proceeding from the first iteration at the top to the last iteration at the bottom. In the plots of the posterior on $f$ and $g$, "x" denotes infeasible points and a square denotes feasible points. The constraint threshold (0) is plotted as a red line. The red dot is the point newly sampled in the current iteration. 2-OPT-C explores potentially infeasible points more aggressively and discovers a better feasible point.

Before presenting out non-myopic acquisition function, 2-OPT-C, we give an intuition for why being non-myopic is even more important in constrained problems than it is in unconstrained ones. Suppose the constraint function value of the (unobserved) best feasible point is equal or close to the constraint threshold. To find this point with as few function evaluations as possible, the algorithms need to efficiently learn the feasible region's boundaries. This happens most quickly when evaluating points close to this boundary. Since the boundary is uncertain, this requires an algorithm to be willing to evaluate points that may be infeasible. However, the value of such points to myopic CBO algorithms like EIC is significantly reduced because the probability of infeasibility is high — typically near 50%, lowering the $\text{PF}(x)$ term in (1). Thus, myopic CBO algorithms may insufficiently explore the boundary, instead sticking well inside the feasible region.

Figure 1 illustrates this with an example. On the first iteration, EIC evaluates a point in a region around an existing feasible point that is likely feasible, getting a small one-step improvement but little information on $f$ and $g$ for future evaluations. However, 2-OPT-C (defined below) explores more

aggressively. The point it evaluates is not feasible but provides a significant amount of information about the boundary between the feasible and infeasible region. As a result, after progressing two steps, 2-OPT-C locates a new feasible point ($g(x) = -0.02$) with a much lower objective function value while EIC remains stuck near the previous feasible point.

### 3.2  Definition of 2-OPT-C

We define our two-step optimal acquisition function for CBO, called 2-OPT-C, for both sequential and batch settings, and then discuss its computational optimization in §4. As in other CBO methods, we sequentially evaluate f and g next at the point that maximizes our acquisition function. This point is optimal to measure now assuming we will be able to measure one additional point before being awarded the maximum value found over the two new measurements and in our previously collected data. In other words, 2-OPT-C looks two steps ahead instead of the one-step ahead considered by myopic methods. For clarity, we assume a single constraint and treat multiple constraints in the supplement.

In the batch setting, we find the batch of $q$ points to measure next assuming we will then measure one additional point before being awarded the best value found. We assume only one additional point rather than a full additional batch for computational tractability. We argue that even the single additional point allows 2-OPT-C to think non-myopically and to value information about the feasible region that will only be acted on later.

We first define notation. We use subscripts $\{0, 1, 2\}$ to denote stages. Stage 0 is the current stage when we will evaluate a batch of points. In stage 1, we will evaluate one additional point based on the results of stage 0 and then in stage 2 will be judged based on the best of these points evaluated. We put independent GP priors on $f$ and $g$. Let $D$ be the collection of data points that we have observed so far. Then given the information in $D$ and their corresponding objective $f(D)$ and constraint $g(D)$ values, the posterior distribution on the objective $f$ is a Gaussian process with a mean function $\mu_0$ and kernel $K_0$. Similarly, the posterior distribution on the constraint $g$ is also a Gaussian process with a mean function $\mu_0^c$ and kernel $K_0^c$. $\mathbb{E}_0$ denotes taking the expectation with respect to the posterior distributions given $D$.

Let $f_0^*$ be the best evaluated point satisfying the constraint so far, i.e. $f_0^* = \min_{x \in D, g(x) \le 0} f(x)$. Let $X_1$ be the set of $q$ candidate points that we consider evaluating at the first stage and let $Y_f = \{f(x) : x \in X_1\}$ and $Y_g = \{g(x) : x \in X_1\}$ be the sets of corresponding objective function values and constraint values respectively. We let $\mu_1, \mu_1^c$ and $K_1, K_1^c$ denote the mean function and kernel for the posterior distributions of $f$ and $g$ respectively given $D$ and $X_1$. Let $\sigma_1(x) = \sqrt{K_1(x, x)}$ and $\sigma_1^c(x) = \sqrt{K_1^c(x, x)}$. Let $\mathbb{E}_1$ indicate the expectation with respect to the corresponding Gaussian processes given both $D$ and $X_1$. Finally, we use $x_2$ to denote a single point to be evaluated in the second stage, based on the results of the first.

Let $f_1^*$ and $f_2^*$ be the best evaluated feasible point by the end of the first and second stage respectively, $f_1^* = \min\{f_0^*, \min_{x \in X_1, g(x) \le 0} f(x)\}$ and $f_2^* = \min\{f_1^*, f(x_2)\}$ if $g(x_2) \le 0$ and $f_2^* = f_1^*$ if not.

Following the principle of dynamic programming, our goal is to choose $X_1$ to minimize the overall expected objective $\mathbb{E}_1(f_2^*)$ . This is done under the assumption that $x_2$ will be chosen optimally leveraging observations of $X_1$. Equivalently, our goal is to maximize

$$\mathbb{E}_0 \left[ \max_{x_2} [f_0^* - f_2^*] \right] = \mathbb{E}_0 \left[ f_0^* - f_1^* + \max_{x_2} \mathbb{E}_1[f_1^* - f_2^*] \right],$$

over $X_1$ chosen in the first stage. $\mathbb{E}_1[f_1^* - f_2^*]$ depends implicitly on the information obtained from $X_1$, which is $Y_f$ and $Y_g$, and is included in the posterior over which $\mathbb{E}_1$ is taken.

Thus, we define the constrained two-step acquisition function:

$$\text{2-OPT-C}(X_1) := \mathbb{E}_0 \left[ f_0^* - f_1^* + \max_{x_2 \in A(\delta)} \mathbb{E}_1 [f_1^* - f_2^*] \right],$$

where $A(\delta)$ is a compact subset of $A$ consisting of points at least $\delta$ away from sampled points in $D \cup X_1$. With $\delta = 0$, $A = A(\delta)$. We introduce the parameter $\delta \ge 0$ purely to overcome a technical hurdle in our theoretical analysis: that the standard deviation of the posterior distribution is not smooth at sampled points. We believe $\delta$ can be set to 0 in practice. Indeed, the theoretical analysis (Theorem 1) allows setting $\delta$ at any arbitrary small positive value.

For use in §4, we derive a more directly computable expression for 2-OPT-C$(X_1)$. We rewrite

$$\text{2-OPT-C}(X_1) = \mathbb{E}_0 \left[ \max_{x_2 \in A(\delta)} [f_0^* - f_1^* + \mathbb{E}_1[f_1^* - f_2^*]] \right] = \mathbb{E}_0 \left[ \max_{x_2 \in A(\delta)} \alpha(X_1, x_2, Y) \right],$$

where $Y = (Y_f, Y_g) \sim p(y; X_1)$ and $p(y; X_1)$ is the distribution of $Y$ given $f(D)$ and $g(D)$, specified explicitly as

$$p(y; X_1) = \mathcal{N} \left( \begin{bmatrix} \mu_0(X_1) \\ \mu_0^c(X_1) \end{bmatrix}, \begin{bmatrix} K_0(X_1, X_1) & 0 \\ 0 & K_0^c(X_1, X_1) \end{bmatrix} \right). \tag{3}$$

Then, $\alpha(X_1, x_2, Y)$ can be written in closed form:

$$\alpha(X_1, x_2, Y) = f_0^* - f_1^* + \text{EI}(f_1^* - \mu_1(x_2), \sigma_1(x_2)^2) \cdot \text{PF}(\mu_1^c(x_2), (\sigma_1^c(x_2))^2),$$

where $\text{EI}(m, v) = m\Phi(m/\sqrt{v}) + \sqrt{v}\varphi(m/\sqrt{v})$ and $\text{PF}(m^c, v^c) = \Phi\left(-m^c/\sqrt{v^c}\right)$.

# 4  Optimizing 2-OPT-C Using Likelihood Ratios

Evaluating 2-OPT-C$(X_1)$ requires performing a simulation where each replication samples $Y$ and then evaluates $\max_{x_2} \in A(\delta)\alpha(X_1, x_2, Y)$. Averaging these replications gives a Monte Carlo estimate of 2-OPT-C$(X_1)$. Optimizing 2-OPT-C$(X_1)$ using such Monte Carlo estimates is difficult because of noise from simulation and because there is not a straightforward way to obtain derivatives.

Indeed, we show in the supplement that the two widely used approaches to efficiently optimizing Monte Carlo acquisition functions, IPA and SAA, fail to optimize 2-OPT-C well because constraints cause discontinuities in the surface they sample using the reparameterization trick.

Here, we develop a novel approach for optimizing Monte Carlo acquisition functions like 2-OPT-C without using the reparameterization trick, overcoming the challenges created by these discontinuities. This approach uses the likelihood ratio method to derive an unbiased estimator of the gradient for 2-OPT-C. We then use this novel estimator in multistart stochastic gradient ascent to maximize 2-OPT-C. (For details on multistart stochastic gradient ascent for maximizing a Monte Carlo acquisition function, see [34].) To the best of our knowledge, we are the first to demonstrate the benefits of the likelihood ratio method for acquisition function gradient estimation in BO.

## 4.1  Background on the Likelihood Ratio Method

We first give background on the likelihood ratio method using generic notation before describing how we use it in our setting. Given a generic random variable $\theta(x)$ whose distribution depends on a control vector $x$ with density $p(\theta; x)$ and a function $V(x, \theta(x))$, our goal is to solve $\max_x \mathbb{E}[V(x, \theta(x))]$. To do this, we estimate the gradient of $\mathbb{E}[V(x, \theta(x))]$ for use within multistart stochastic gradient.

To provide this gradient estimator, we first choose a density $\tilde{p}(\theta)$ that does not depend on $x$ and for which $\{\theta : p(\theta, x) > 0\} \subseteq \{\theta : \tilde{p}(\theta) > 0\}$ for all $x$. Using this density, we construct the likelihood ratio, $L(\theta; x) = p(\theta; x)/\tilde{p}(\theta)$. We then have that

$$\mathbb{E}[V(x, \theta(x))] = \int V(x, \theta) \, p(\theta; x) \, d\theta = \int V(x, \theta) \, L(\theta; x)\tilde{p}(\theta) \, d\theta.$$

This is referred to as importance sampling and $\tilde{p}$ is referred to as the importance sampling distribution.

The likelihood ratio method uses this expression to construct a gradient estimator. Under regularity conditions [35], we can exchange the gradient operator and integration,

$$\nabla_x \mathbb{E}[V(x, \theta(x))] = \nabla_x \int V(x, \theta) \, L(\theta; x)\tilde{p}(\theta) \, d\theta = \int \nabla_x V(x, \theta) \, L(\theta; x)\tilde{p}(\theta) \, d\theta.$$

From this we can create an unbiased estimator of the gradient of $\mathbb{E}[V(x, \theta(x))]$ by sampling $\theta$ from the density $\tilde{p}(\theta)$ and returning as our estimator $\nabla_x V(x, \theta)L(\theta; x)$.

A natural choice for the importance sampling distribution suggested in [24] is to take $\tilde{p}(\theta) = p(\theta, \tilde{x})$ when estimating $\nabla_x \mathbb{E}[V(x, \theta(x))]$ at $x = \tilde{x}$. We make this choice when designing an unbiased gradient estimator for 2-OPT-C. We discuss this in detail later in §4.2.

## 4.2 Optimizing 2-OPT-C

To apply the likelihood ratio method to develop a gradient estimator for 2-OPT-C, we begin by rewriting the expression to be differentiated using importance sampling.

Mapping the notation of our generic discussion of importance sampling onto our specific problem, our control $x$ is the batch of points $X_1$, our random variable $\theta$ is $Y$ (whose distribution depends on $X_1$), and $V(x, \theta)$ is $\max_{x_2} \alpha(X_1, x_2, Y)$. $Y$ is a multivariate normal random variable whose density we write $p(y; X_1)$, noting that its mean and covariance matrix are determined by $X_1$. We let $p(y; \tilde{X}_1)$ be our importance sampling density for $Y$, for some fixed point $\tilde{X}_1$. $\tilde{X}_1$ is arbitrary for now but is specified below. Our likelihood ratio is then $L(y; X_1, \tilde{X}_1) = p(y; X_1)/p(y; \tilde{X}_1)$.

As discussed above, the key step in the likelihood ratio method is to interchange the integral and gradient operator. This is justified in our setting by Theorem 1 below. The proof is provided in the supplement. As a result, we have

$$\nabla_{X_1} \text{2-OPT-C}(X_1) = \int \Gamma(X_1, \tilde{X}_1, y) \, p(y; \tilde{X}_1) dy, \tag{4}$$

where

$$\Gamma(X_1, \tilde{X}_1, y) := \nabla_{X_1} \left[ \max_{x_2 \in A(\delta)} \alpha(X_1, x_2, y) L(y; X_1, \tilde{X}_1) \right]$$

$$= \left[ \nabla_{X_1} \alpha(X_1, x_2^*, Y) \right] L(y; X_1, \tilde{X}_1) + \alpha(X_1, x_2^*, Y) \left[ \nabla_{X_1} p(y; X_1) \right] / p(y; \tilde{X}_1)$$

with $x_2^* \in \arg\max_{x_2 \in A(\delta)} \alpha(X_1, x_2, Y)$. The last equality is by the envelope theorem [36], i.e., $\max_{x_2 \in A(\delta)} \alpha(X_1, x_2, Y)$ can be differentiated with respect to $X_1$ by first optimizing over $x_2$ given $X_1$, then holding $x_2^*$ fixed while differentiating with respect to $X_1$. From now on, we will drop the subscript of the differential operator $\nabla$ for simplicity.

By (4), $\Gamma(X_1, \tilde{X}_1, Y)$ is an unbiased estimator of $\nabla \text{2-OPT-C}(X_1)$ when $Y$ is drawn according to $p(y; \tilde{X}_1)$. With this stochastic gradient estimator, we then can use stochastic gradient ascent [37] with multiple restarts to find a collection of stationary values for $X_1$. Then we use simulation to evaluate 2-OPT-C$(X_1)$ at these values and select the one with the highest estimated 2-OPT-C. This then provides a computationally efficient algorithm for optimizing 2-OPT-C. Pseudocode for using 2-OPT-C is provided in the supplement.

As discussed above, the key step to developing our unbiased estimator is to interchange the integral and gradient operator, which is justified by Theorem 1 below. The proof is provided in the supplement. Although Theorem 1 assumes $\delta > 0$, in practice, we choose $\delta = 0$ in our stochastic gradient estimator.

**Theorem 1.** *We assume:*

1. *The prior on the objective function $f$ is a Gaussian Process $f \sim GP(\mu_f(x), K_f(x, x'))$, and the prior on the constraints $g$ is another Gaussian Process $g \sim GP(\mu_g(x), K_g(x, x'))$. These two Gaussian processes are independent.*

2. *$\mu_f$ and $\mu_g$ is continuously differentiable with $x$.*

3. *$K_f(x, x')$ and $K_g(x, x')$ is continuously differentiable with $x$ and $x'$.*

4. *Given $n$ different points $X = (x_1, x_2, ..., x_n)$, the matrix $K_f(X, X)$ and $K_g(X, X)$ are of full rank.*

*Then 2-OPT-C$(X_1)$'s partial derivatives exist almost everywhere for any $\delta > 0$. When 2-OPT-C$(X_1)$ is differentiable,*

$$\nabla \text{2-OPT-C}(X_1) = \int \Gamma(X_1, \tilde{X}_1, y) p(y; \tilde{X}_1) dy$$

**Choice of Importance Sampling Distribution** We have constructed an estimator of the gradient $\nabla \text{2-OPT-C}$ at $X_1$. This estimator was constructed using an importance sampling distribution parameterized by $\tilde{X}_1$. We are free to choose this $\tilde{X}_1$ as we wish. We recommend setting $\tilde{X}_1$ equal

to $X_1$, as our gradient estimator takes a particularly simple form and offers robust performance. In particular, when $\widetilde{X}_1 = X_1$, $L(y; X_1, X_1) = 1$, so our gradient estimator is

$$\Gamma(X_1, X_1, y) := \alpha(X_1, x_2^*, y) \left[\nabla p(y; X_1)\right] / p(y; X_1) + \left[\nabla \alpha(X_1, x_2^*, y)\right],$$

where $x_2^* \in \arg\max_{x_2} \alpha(X_1, x_2, y)$.

## 5 Numerical Experiments

This section numerically investigates our algorithms on problems widely used as benchmarks in the constrained BO literature. Benchmarks demonstrate that 2-OPT-C usually provides significant improvements in query efficiency over state-of-the-art methods. In the main paper, we focus on query efficiency for the sequential setting. The supplement includes additional experiments and discussions of batch evaluations and 2-OPT-C's computational overhead.

The benchmark problems include three synthetic problems from [1], named P1, P2, and P3, and two real-world problems, portfolio optimization and robot pushing. Detailed descriptions are in the supplement. We use these five problems in comparisons with myopic methods in §5.2. In §5.3 we compare with the non-myopic method from [1]. Since code for [1] was not available, we compare 2-OPT-C against the results previously published in that paper, consisting only of P1, P2, and P3.

### 5.1 Experiment Setup

**Evaluation Metrics** We follow [11] in our evaluation methodology. Along with each evaluation $n$ each algorithm makes a "recomendation", which is the point we would evaluate if it were our last evaluation before being scored by the best point evaluated. From this recommended point, we compute a score $f_n^{**}$ for this algorithm in this timestep $n$. If the recommended point is feasible, then $f_n^{**}$ is its objective function value. If not, $f_n^{**}$ is the best observed feasible value so far before the recommended point. Following [11], the point recommended is the one with the lowest posterior mean objective value, among those whose probability of satisfying each constraint is 0.975 or better. We then report the utility gap, $\epsilon_n = |f_n^{**} - f^*|$, which is the difference between this score and the global constrained optimum $f^*$. We report the log10 median utility gaps for P1, P2, P3, and the robot pushing problem in the main paper. Mean utility gap results are provided in the supplement. In the portfolio optimization problem, the optimum is unknown so we report the mean annualized return rate instead of the utility gap.

**Setup for §5.2** The 2-OPT-C implementation uses GPs with a constant zero-mean prior and ARD square-exponential kernels for both objectives and constraints. GP hyperparameters are obtained by maximizing the marginal likelihood using GPy [38]. SAA+CMA_ES is implemented similarly to 2-OPT-C and all the hyperparameters are the same or obtained in the same way as 2-OPT-C.

For all three synthetic problems (P1, P2, and P3), we run 150 experiment replications for all algorithms. For the two real-world problems (portfolio optimization and robot pushing), we run 50 experiment replications. For the initialization of each experiment, we randomly sample three points with at least one feasible point from a Latin hypercube design. We run $N = 40$ function evaluations for P1 and P2, $N = 60$ for P3, $N = 30$ for portfolio optimization problem, and $N = 50$ for robot pushing problem. We use batch size of 1 for all five experiments.

**Setup for §5.3** The setup for §5.3 is nearly the same as for §5.2, with three key differences. These arose from the need to replicate the experimental setup from [1].

First, we run 500 experiment replications. Second, rather than 3 initial points per problem, we use 1. Third, the evaluation method differs slightly. If the recommended point is infeasible, then rather than setting $f_n^{**}$ to the value of the best previously observed feasible point, we set it to the maximum of the objective function over all points in the domain, thus enforcing a substantial penalty.

### 5.2 Comparison with Myopic Methods

Figure 2 first compares the myopic methods ADMMBO [12], BO_Slack [15], NEI [16], EIC [9, 8] and PESC [11] on three synthetic problems. In general, 2-OPT-C offers significantly lower median

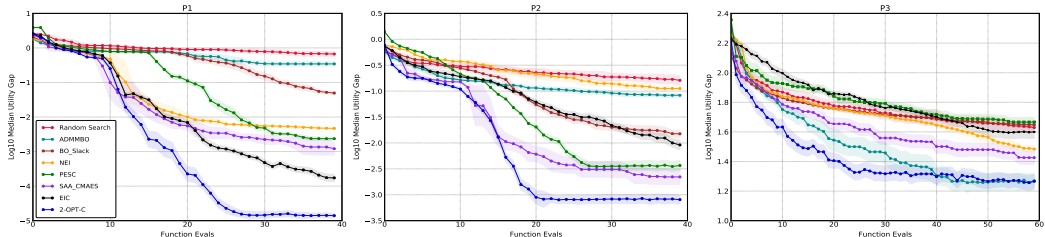

Figure 2: Log10 median utility gap of Random Search, ADMMBO, BO_Slack, NEI, EIC, PESC, SAA_CMAES, and 2-OPT-C with 95% confidence intervals for problems P1, P2, and P3.

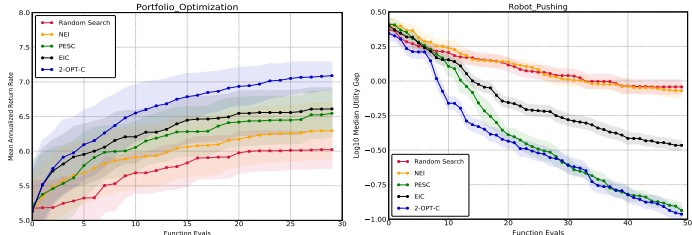

Figure 3: Real-world experiment results on Random Search, NEI, EIC, PESC, and 2-OPT-C. We report the mean annualized return rate for portfolio optimization problem and log10 median utility gap for robot pushing problem.

utility gaps than the myopic methods across all three problems. For example, in P1, 2-OPT-C quickly converges to a utility gap $10^{-5}$ (in 27 evaluations), while the best of the other benchmarks (EIC) has a utility gap of $10^{-3}$ at this many evaluations. There is only one method and problem, ADMMBO on problem P3, in which a myopic method achieves an optimality gap comparable to 2-OPT-C, but 2-OPT-C reaches this optimality gap in fewer iterations.

Figure 2 additionally compares with an SAA implementation of 2-OPT-C that uses CMA-ES [39] to optimize the SAA to the acquisition function on P1, P2, and P3, denoted as SAA_CMAES. We see that SAA_CMAES generally perform better than myopic policies on all three problems (except EIC on P1), supporting the value of two-step lookahead for CBO over myopic approaches. However, due to the discontinuity introduced by SAA, it underperforms 2-OPT-C on all three synthetic problems. It also underperforms EIC in P1 and ADMMBO in P3. In addition, it requires substantially more computation. This is because, as we note in the supplement, discontinuities in the SAA to the acquisition function make it extremely difficult to optimize. We provide additional experiments to illustrate this in the supplement.

Figure 3 compares 2-OPT-C with the mostly widely-used myopic methods, NEI, PESC, and EIC, on real-world problems. As in the synthetic problems, 2-OPT-C outperforms the competing benchmarks. In portfolio optimization, 2-OPT-C provides roughly $0.5\%$ more annualized return than the best myopic method (EIC). In robot pushing, 2-OPT-C outperforms EIC and NEI over the full range of function evaluations; PESC eventually matches 2-OPT-C but there is a range of function evaluations where 2-OPT-C performs strictly better.

### 5.3 Comparison with the Non-myopic Method

We compare 2-OPT-C with the non-myopic rollout algorithm of [1] on the three synthetic problems based on the results reported in their paper. We use the terms Rollout-1, Rollout-2, and Rollout-3 to denote this algorithm with horizons of 1, 2, and 3, respectively.

To compare with the rollout algorithm, we extract experimental data from [1] and replicate their experimental setup, adding 2-OPT-C's results to their Figures 2 and 3.

Figure 4 reports the results of these comparisons. It shows that 2-OPT-C provides query efficiency that is at least as good as the non-myopic rollout methods with different horizons on all three problems and is sometimes substantially better. In detail, for P1, 2-OPT-C increases solution quality by more than a factor of 2 ($10^{-4.59}/10^{-4.92}$) compared to the best rollout strategies, i.e., Rollout-1. For P3,

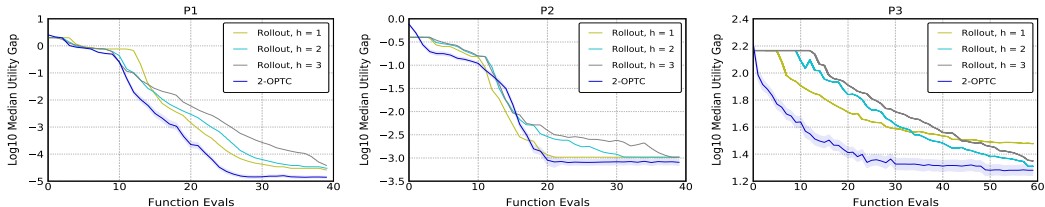

Figure 4: Log10 median utility gap of 2-OPT-C and the existing non-myopic constrained BO algorithm with different rollout horizons on P1, P2, and P3. The three lines of the rollout algorithm are extracted from Figure 3 in [1]. Additional information is provided in Table 1 in the supplement.

2-OPT-C is able to improve solution quality over the best competing rollout method by roughly a factor 2 when the evaluation budget is tight (after 30 function evaluations).

2-OPT-C also uses less computation than the rollout algorithm [1]. Since the code from [1] is not publicly available, we refer to discussion in [21] of the computational time of an unconstrained version of the rollout algorithm, which we understand from private communication has a similar cost to [1]. This states that the unconstrained rollout strategies require from 10 minutes to 1 hour to evaluate a single point even on a low-dimensional problem. This may be even larger in constrained settings, since we need a separate GP to model the constraint function. 2-OPT-C is more computationally efficient. On P3, for instance, 2-OPT-C optimizes a batch of 5 points in 25-30 minutes.

Rollout-1 is actually attempting to optimize the same theoretical acquisition function used by 2-OPT-C. Indeed, [1] uses EIC as its rollout strategy, which is optimal over a horizon of $h = 1$. Rollout-1 is then trying to identify the point to evaluate now that would be optimal if one additional point were evaluated after using EIC. If this point could be found exactly, it would be 2-step optimal.

Despite attempting to optimize an acquisition function that is conceptually the same as 2-OPT-C, the performance of Rollout-1 is substantially worse (the utility gap is roughly 2x larger over all budgets), despite requiring more than double the computation. This is because [1] optimizes this two-step acquisition function without the type of derivative estimates provided by our likelihood ratio method.

## 6 Conclusion

This paper presents a non-myopic CBO algorithm, supporting both batch and sequential settings, that substantially improves both query efficiency and computation time over the one previous method focused on this class of problems for both sequential and batch settings.

While our method offers significant improvements in query efficiency compared to the state-of-the-art that more than offset its computational expense, there are likely more opportunities for reducing its computational cost. This, as well as a further theoretical investigation regarding our method providing such significant improvements in query efficiency compared to other methods, presents valuable directions for future work.

Faster constrained optimization of time-consuming-to-evaluate functions with our methodology enables creation of new engineering systems, better supervised learning methods, and lower-cost business operations. While we believe technological innovation is on balance good, new technology created using our methods also has the potential for negative effects. Understanding the societal effects of improved optimization capabilities is an important area for future research.

## 7 Acknowledgements

The authors were partially supported by AFOSR FA9550-19-1-0283 and FA9550-20-1-0351.

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
