# Two-step Lookahead Bayesian Optimization with Inequality Constraints Supplement

The supplement is organized as follows.

- §A demonstrates that standard methods for efficient gradient-based optimization of Monte Carlo acquisition functions, IPA and SAA, fail for optimizing 2-OPT-C because of discontinuities that arise when using the reparameterization trick.

- §B presents detailed descriptions of the experiment problems from the main paper, and additional experiment results for the batch setting and computational overhead.

- §C presents a more general version of 2-OPT-C that handles multiple constraints.

- §D provides detailed pseudocode for optimizing 2-OPT-C using SGD and our likelihood ratio estimator.

- §E proves Theorem 1 in the main paper.

- For completeness, §F presents a more detailed treatment of batch constrained expected improvement defined in §2.3 of the main paper.

## A  Discontinuities Hinder Optimization via the Reparameterization Trick

Using 2-OPT-C for CBO requires us to evaluate and optimize the 2-OPT-C acquisition function. This acquisition function cannot be evaluated exactly, and must instead be evaluated via Monte Carlo. Derivative-free optimization of such Monte Carlo acquisition functions, e.g., using CMA-ES, is typically computationally expensive and requires a substantial number of acquisition function simulation replications to optimize well.

In response to this challenge, there are two widely-used special-purpose methods for optimizing Monte Carlo acquisition functions: infinitesimal perturbation analysis (IPA), and sample average approximation (SAA). IPA efficiently estimates the gradient of acquisition function and then uses this gradient estimate within multistart stochastic gradient ascent. SAA uses Monte Carlo samples to create a deterministic and differentiable approximation to the acquisition function, which it then optimizes using derivative-based deterministic optimization. The first papers known by the authors using IPA and SAA for optimizing BO acquisition functions are [1] (which was available earlier on arxiv [2]) and [3] respectively.

As we explain in detail below, both IPA and SAA rely on a Monte Carlo approximation to the acquisition function simulated using the reparameterization trick. In particular, they rely on this approximation being smooth in the input to the acquisition function. However, in *constrained* BO and as we argued in the main paper, this approximation is not smooth and indeed has discontinuities. This prevents the efficient use of IPA and SAA for optimizing 2-OPT-C and necessitates the development of our new likelihood ratio approach.

In this section, we give background on IPA and SAA and then describe in detail these discontinuities and why they prevent the use of IPA and SAA for optimizing 2-OPT-C.

### A.1  Background on IPA and SAA

For our discussion of IPA and SAA we define generic notation. Let $V(x, \theta)$ be a function of two variables, one that will be a control vector $x$ and the other that will be a random input $\theta$. Critically, for both IPA and SAA, the distribution of $\theta$ does not depend on $x$. This contrasts with the likelihood ratio method described in the main paper, in which the distribution of $\theta$ *can* depend on $x$.

Here, we will discuss the generic use of IPA and SAA for solving $\max_x E[V(x, \theta)]$ where the expectation is over $\theta$. In the next section, where we focus on the difficulties arising when optimizing 2-OPT-C using IPA and SAA, $x$ will be replaced by the batch of points that we consider evaluating in the first stage and $\theta$ will be a vector of two standard normal random variables used in the reparameterization trick.

**IPA** We first review the concept of IPA, which is a method for estimating the gradient $\nabla_x E[V(x, \theta)]$. Once this gradient is estimated, it can be used within multistart stochastic gradient ascent.

IPA estimates this gradient as follows. Under some regularity conditions [4], the gradient operator and the expectation can be swapped,

$$\nabla_x E[V(x, \theta)] = E[\nabla_x V(x, \theta)]. \tag{1}$$

One can estimate the the right-hand side using Monte Carlo. This is accomplished by generating i.i.d. samples $\theta_i$ from the distribution of $\theta$ and then using the estimate $\frac{1}{N} \sum_{i=1}^{N} \nabla_x V(x, \theta_i)$ This is an unbiased estimator of the right-hand side of (1). When the equality in (1) holds, this is also an unbiased estimator of $\nabla_x \mathbb{E}[V(x, \theta)]$.

Unfortunately, however, (1) does not always hold. To illustrate, suppose $\theta$ is a standard normal random variable, $\mu(x)$ and $\sigma(x)$ are two continuous functions, and $V(x, \theta) = 1\{\mu(x) + \sigma(x)\theta \leq 0\}$. Observe that $V(x, \theta)$ can be discontinuous in $x$ for any given value of $\theta$.

Then the left-hand side of (1) is $\frac{\mathrm{d}}{\mathrm{d}x} E[V(x, \theta)] = \frac{\mathrm{d}}{\mathrm{d}x} \Phi(-\mu(x)/\sigma(x))$ where $\Phi$ is the standard normal cumulative distribution function. The right-hand side of (1) is $E[\frac{\mathrm{d}}{\mathrm{d}x} V(x, \theta)] = 0$, since $\frac{\mathrm{d}}{\mathrm{d}x} V(x, \theta)$ is 0 at all $\theta$ except where $\theta = -\mu(x)/\sigma(x)$, and this $\theta$ occurs with probability 0. Thus, in this example, the left-hand side of (1) is different from the right-hand side.

The essential difficulty in this example is the discontinuity in $V(x, \theta)$ as we vary $x$. This allows the derivative of $V(x, \theta)$ to be 0 for a set of $\theta$ with probability 1 while the expected value of $V(x, \theta)$ is nevertheless non-zero. This failure of IPA caused by $V(x, \theta)$ being discontinuous in $x$ is similar to its failure in our setting. While we have a closed form analytic expression for $\mathbb{E}[V(x, \theta)]$ in this simple example, in situations like 2-OPT-C where we do not, this issue prevents the use of IPA for estimating derivatives.

**SAA** We now briefly review SAA. As above, we want to maximize $\mathbb{E}[V(x, \theta)]$. In SAA, we first sample $\theta_1, \theta_2, \ldots, \theta_M$ (usually referred as base samples) i.i.d. from the distribution of $\theta$. Then to optimize $\mathbb{E}[V(x, \theta)]$, SAA applies a deterministic gradient-based optimization algorithm to optimize $\widehat{V}_M(x)$, where $\widehat{V}_M(x) = \frac{1}{M} \sum_{m=1}^{M} V(x, \theta_m)$. More details about SAA can be found in [5].

As we demonstrate later in the section, writing our acquisition function 2-OPT-C$(X_1)$ in this form using the reparameterization trick results in a $V_M(x)$ that is discontinuous in $x$, where the number of discontinuities grows with $n$. This can also be seen by applying SAA to the example above, $V(x, \theta) = 1\{\mu(x) + \sigma(x)\theta \leq 0\}$. In this example, $\widehat{V}_M(x) = \frac{1}{M} \sum_{m=1}^{M} 1\{\mu(x) + \sigma(x)\theta_m \leq 0\}$, which has jumps of size $1/M$ at $x$ where $\mu(x) + \sigma(x)\theta_m = 0$ for any $m$. Discontinuous functions are difficult to optimize well, especially when there are many discontinuities. Moreover, seeking to improve the accuracy of $\widehat{V}_M(x)$ by increasing the number of samples $M$ also increases the number of points $x$ with a discontinuity. This makes it difficult to use SAA for such discontinuous functions.

## A.2  Discontinuities with the Reparameterization Trick

We now show how discontinuities arising from the reparameterization trick prevent the use of IPA and SAA for optimizing 2-OPT-C. Using the reparameterization trick, we first write $Y_f$ and $Y_g$ as deterministic functions of normal random variables: $Y_f = \mu_0(X_1) + R_0(X_1)Z_f$, $Y_g = \mu_0^c(X_1) + R_0^c(X_1)Z_g$, where $Z_f, Z_g$ are two $q$-dimensional independent standard normal random variables and $R_0(X_1), R_0^c(X_1)$ are the Cholesky decompositions of $K_0(X_1, X_1)$ and $K_0^c(X_1, X_1)$.

To optimize 2-OPT-C, we would replace the values of $Y_f$ and $Y_g$ in the definition of 2-OPT-C$(X_1)$ with these expressions. We focus on the term $\mathbb{E}_0[f_0^* - f_1^*]$, since this is where the difficulty lies, and focus on the non-batch case for simplicity. We have:

$$\mathbb{E}_0[f_0^* - f_1^*] = \mathbb{E}_0[(f_0^* - f(X_1))^+ \cdot \mathbb{1}\{g(X_1) \leq 0\}]$$
$$= \mathbb{E}_0[(f_0^* - \mu_0(X_1) - R_0(X_1)Z_f)^+ \cdot \mathbb{1}\{\mu_0^c(X_1) + R_0^c(X_1)Z_g \leq 0\}].$$

The key difficulty for both IPA and SAA arises because the term $\mathbb{1}\{\mu_0^c(X_1) + R_0^c(X_1)Z_g \leq 0\}$ has a discontinuity as we vary $X_1$, at points where $\mu_0^c(X_1) + R_0^c(X_1)Z_g$ is 0.

We describe this difficulty, beginning with IPA. To use IPA to estimate the gradient of 2-OPT-C in support of a gradient-based optimization method, one would first sample $Z_f$ and $Z_g$ and then use the gradient with respect to $X_1$ of the expression inside the expectation, $V(X_1, \theta) = (f_0^* - \mu_0(X_1) - R_0(X_1)Z_f)^+ \cdot \mathbb{1}\{\mu_0^c(X_1) + R_0^c(X_1)Z_g \leq 0\}$, as an estimator of $\nabla_{X_1} \mathbb{E}_0[f_0^* - f_1^*] = \nabla_{X_1} \mathbb{E}_0[V(X_1, \theta)]$, where $\theta = (Z_f, Z_g)$.

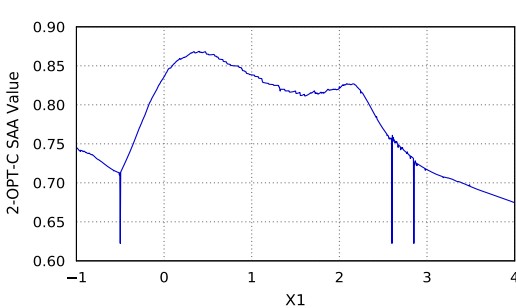

Figure 1: SAA approximation to 2-OPT-C$(X_1)$ using 256 base samples, plotting the approximation vs. $X_1$ in a 1-d problem with one constraint. Computation grows as dimensions, constraints, or samples are added. As sample size increases, jumps shrink in size but grow in number. The three large drops are due to ill-conditioned matrices near previously-evaluated points, while our focus is on the many small jumps, which can be seen better after zooming in. The surface's roughness prevents efficient optimization with more than a few dimensions or constraints.

Unfortunately, however, the term $\mathbb{1}\{\mu_0^c(X_1) + R_0^c(X_1)Z_g \leq 0\}$ is discontinuous, taking two different constant values for different values of $X_1$. While modifying $X_1$ affects the probability of feasibility, the derivative of this term with respect to $X_1$ will be $0$ (as long as $X_1$ avoids the boundary between feasible and infeasible regions). This causes the gradient estimator to be biased, preventing its use for optimizing the acquisition function.

The discontinuity in $\mathbb{1}\{\mu_0^c(X_1) + R_0^c(X_1)Z_g \leq 0\}$ also creates a problem for SAA. In our context, to optimize our acquisition function $\mathbb{E}_0[f_0^* - f_1^*]$, SAA would sample standard normal random vectors $Z_{f,m}$ and $Z_{g,m}$ of appropriate dimension and then optimize the following over $X_1$:

$$\frac{1}{M}\sum_{m=1}^{M}\left[(f_0^* - \mu_0(X_1) - R_0(X_1)Z_{f,m})^+ \cdot \mathbb{1}\{\mu_0^c(X_1) + R_0^c(X_1)Z_{g,m} \leq 0\}\right]. \tag{2}$$

Unfortunately, the discontinuity in $\mathbb{1}\{\mu_0^c(X_1) + R_0^c(X_1)Z_g \leq 0\}$ causes (2) to be discontinuous, as shown in Figure 2, making it difficult to optimize. To combat these discontinuities, one approach is to average over a larger number of samples $M$, paying a higher computational cost. This shrinks the jumps but increases their number. Many jumps (even small ones) make gradient-based methods fail. Instead, SAA for constrained problems requires derivative-free methods. Derivative-free methods are much slower for acquisition-function optimization than derivative-based ones ([3] Appendix F, [1] Figs 3 & 4), especially for the problems with higher dimensions or more constraints. In numerical experiments in §5.1, we compare with an SAA-based implementation of two-step lookahead and find that it requires more computation and offers degraded query efficiency compared to our method.

## B   Additional Experiment Details and Results

Here, we present detailed descriptions of the experiment problems from the main paper and additional experimental results. Specifically, we present:

- Details of the benchmark problems from the main paper (§B.1).

- Additional experimental results for the batch setting and investigating the computational overhead associated with acquisition function optimization (§B.2).

- Additional experiments studying the mean utility gap rather than the median reported in the main paper (§B.3).

- Additional experimental results for non-myopic problems (§B.4).

### B.1   Benchmark Problems

Here we give details about the problems used as benchmarks in the main paper.

**P1** : The first synthetic problem is a multi-modal objective with single constraint introduced in [6], which minimizes the objective function $f$ on the design space $A = [0,6]^2$ subject to one constraint:

$$f(x) = \cos(2x_1)\cos(x_2) + \sin(x_1),$$
$$g(x) = \cos(x_1)\cos(x_2) - \sin(x_1)\sin(x_2) + 0.5.$$

**P2** : The second synthetic problem has a linear objective with multiple constraints and is discussed in [7] and [8]. The design space is a unit square $A = [0, 1]^2$ and the objective function and the constraints are given by:

$$f(x) = x_1 + x_2,$$
$$g_1(x) = 0.5 \sin(2\pi(2x_2 - x_1^2)) - x_1 - 2x_2 + 1.5,$$
$$g_2(x) = x_1^2 + x_2^2 - 1.5.$$

**P3** : The third synthetic problem is a multi-modal 4-d objective with a single constraint proposed in [9]. The design space is $A = [-5, 5]^4$. The objective function and the constraint are defined as:

$$f(x) = \frac{1}{2} \sum_{i=1}^{4} (x_i^4 - 16x_i^2 + 5x_i),$$
$$g(x) = -0.5 + \sin(x_1 + 2x_2) - \cos(x_3)\cos(2x_4).$$

**Portfolio Optimization** : The first real-world problem is a portfolio optimization problem based on [10], as studied in a BO context by [11]. In this problem, we optimize an algorithmic trading strategy to maximize a portfolio's mean annualized return rate while constraining the portfolio's risk, as measured by the standard deviation of the return rate. The portfolio simulation uses CVXPortfolio and real-world market data, as described in [10]. A single function evaluation using CVXPortfolio takes between 10 and 15 minutes. The constraint value we choose for this problem is $2(\%)$.

**Robot Pushing** : The second real-world problem a robot pushing problem based on [12], where the robot pushes an object from the origin, i.e., $L_{object} = (0, 0)$, to an unknown target location $L_{target} \in [-5, 5]^2$. The parameters to be optimized are the location of the robot, i.e., $L_{robot} = (x_{robot}, y_{robot})$ and the duration of the push $t \in [1, 30]$. Therefore, the decision variables to optimize are $(x_{robot}, y_{robot}, t)$. The objective is to minimize the distance between the location of the object after being pushed and the target location, namely the $L_2$ norm of $(L_{object \text{ after push}} - L_{target})$. The cost function is the energy used by the robot for pushing the object which is $||L_{object \text{ after push}}|| + \epsilon$, where $\epsilon$ is some noise. The constraint value we use for this problem is 6.

## B.2 Batch Evaluations, Overhead of Acquisition Function Evaluation

Here, we study two additional questions not studied in the main paper: performance in the batch setting; and the overhead required to optimize the acquisition function. The experiment setup is the same as that of §5.2 in the main paper.

While 2-OPT-C is more query efficient, i.e., uses fewer evaluations of the objective function and the constraint than all other methods, it requires more computation time than myopic methods to decide where to evaluate. (It requires less computation than other non-myopic methods, discussed below.) Thus, whether it reduces the *total* computation time required to solve an optimization problem depends on how time-consuming the objective and constraints are to evaluate.

If each evaluation is extremely fast, e.g., milliseconds, then even substantially better query efficiency is not enough to overcome a small elevation in overhead. If each evaluation is slow, however, e.g., minutes or hours, then the time saved via improved query efficiency makes up for a modest increase in overhead. Indeed, BayesOpt is almost always applied in the latter setting, when evaluations are slow.

We present experimental results on one of the example problems from the main paper, P3, showing that 2-OPT-C solves optimization problems more quickly than myopic constrained BayesOpt methods for ranges of objective / constraint evaluation time typical to BayesOpt. We use batch sizes of $q = 1$ (i.e., sequential), $q = 5$ and $q = 10$. All the experiments are run on Amazon Web Services c4.4xlarge instances.

Although the actual benchmark problems are fast to evaluate, we can easily simulate what wall-clock times would result given different amounts of time required to compute the objective function and constraints. We do this by simply adding the actual wall-clock time needed to perform acquisition function optimization to the simulated wall-clock time associated with the number of objective/constraint evaluations performed. We can then calculate the utility gap versus simulated wall-clock time. We do this for four different levels of simulated wall-clock required for an objective / constraint evaluation: 0 minutes; 20 minutes; 40 minutes; and 60 minutes.

Figure 2 shows the results of these experiments, showing total computation time to reach a given mean utility gap under 2-OPT-C with $q = 1$, $q = 5$, and $q = 10$ as compared with myopic BO methods. As the time needed per evaluation for objective / constraint functions varies from 0 to 60 minutes, we plot the total computation time v.s. the log10 utility gap in Figure 2(a-d).

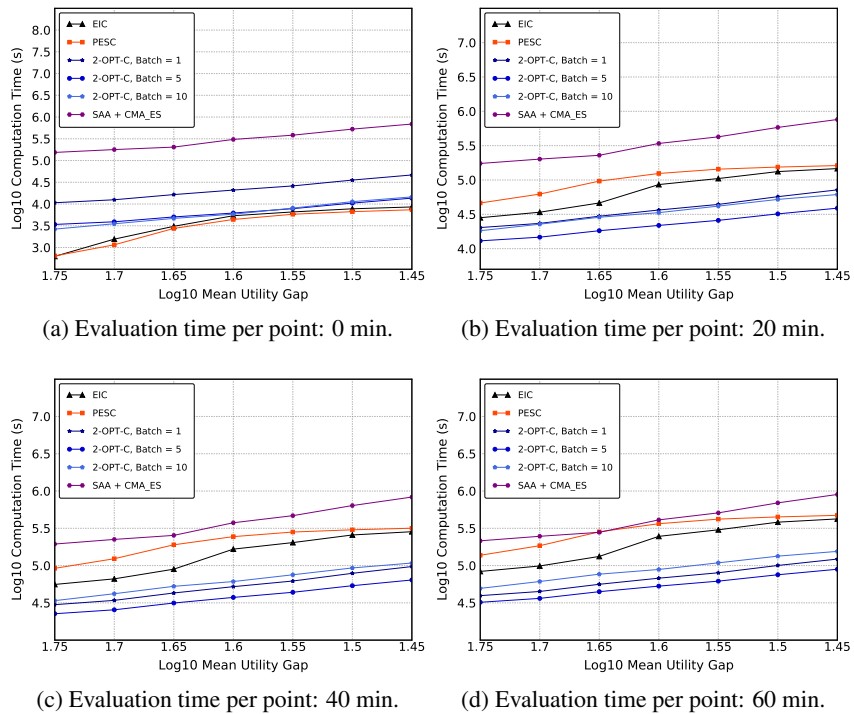

(a) Evaluation time per point: 0 min.      (b) Evaluation time per point: 20 min.

(c) Evaluation time per point: 40 min.      (d) Evaluation time per point: 60 min.

Figure 2: Runtime comparison for EIC, PESC, SAA+CMA_ES and 2-OPT-C. Each point represents the log10 computational time required for the algorithm to achieve a certain level of log10 mean utility gap. The time needed per evaluation for objective / constraint functions ranges from 0 min (Figure a) to 20 min (Figure b), 40 min(Figure c) and 60 min (Figure d).

We make several observations. First, when the objective and constraints are time-consuming-to-evaluate, 2-OPT-C performs much better than the other methods. Although its computational overhead is higher, the time saved by its substantially better query efficiency more than makes up for overhead. When the objective is fast-to-evaluate (0 minutes), 2-OPT-C's computational overhead causes it to underperform other methods. In this regime, however, one would not use Bayesian optimization — even the relatively mild overhead of a method like constrained EI would be undesirable and it would be better to use, e.g., random search or a genetic algorithm.

Second, the batch version of 2-OPT-C is particularly effective. This is for two reasons. First, 2-OPT-C is optimized via a gradient-based method that retains its effectiveness in the higher-dimensional optimization problems created by batch acquisition function optimization. Thus, the overhead required to optimize the acquisition function does not grow substantially as the batch size $q$ grows: optimizing the multiple-point acquisition function for a batch size of $q$ is faster than optimizing the single-point acquisition function $q$ times. This is because the bottleneck in optimizing 2-OPT-C's acquisition function is the "inner" problem, i.e., finding the optimal $x_2^*$ given $X_1$. The time spent solving this inner problem does not grow substantially with $q$. With larger $q$, this overhead is amortized over a larger number of function evaluations. Second, surprisingly, there is little loss in query efficiency associated with moving to the batch setting in this problem, which is evidence that 2-OPT-C effectively optimizes its batch acquisition function. Note that one can use batch 2-OPT-C even when evaluations must be sequential: choose $q$ points to evaluate simultaneously with one large acquisition function optimization, then evaluate them one at a time without updating the GPs until the full batch finishes. When function evaluations are less expensive, this is an appealing way to retain query efficiency while saving computational overhead.

Third, SAA+CMA_ES performs poorly, despite the fact that it is optimizing the same theoretical acquisition function as 2-OPT-C. This points to the value of our novel approach to optimizing this acquisition function. Indeed, although Figure 1 shows that the SAA+CMA_ES approach has better query efficiency compared to myopic methods (EIC and PESC), the discontinuity issue discussed in the main paper causes the computational overhead to be considerably larger than for both myopic methods and 2-OPT-C, while also degrading query efficiency. This in turn causes SAA+CMA_ES to be the worst of the methods in nearly all settings considered.

Fourth, 2-OPT-C does better when one desires a better solution quality (lower mean utility gap). This is because the convergence of myopic methods is slower: the multiplicative gap in the number of evaluations required to reach a given optimality gap grows as the optimality gap grows small.

The above discussion compares 2-OPT-C to the myopic methods EIC and PESC and to SAA+CMA_ES. Here we briefly discuss computation time relative to the other non-myopic method [9]. For 2-OPT-C, under the sequential setting ($q = 1$), it takes roughly 15 to 20 minutes over computational overhead per evaluation. However, [9] is likely to take even more time: a similar code designed for the simpler unconstrained setting requires 10 minutes to 1 hour per iteration, as reported in [13] based on private communication with the authors. Modeling the constraint in a non-myopic way requires a substantially larger scenario space and also doubles the number of GPs to be modeled. This suggests that, in the sequential setting, our method is more efficient than [9]. Moreover, in the batch setting, optimizing 2-OPT-C requires only a modest amount of additional computation per batch as the batch size grows, thus resulting in substantially less overhead per point. For example, choosing a batch size of 5 with 2-OPT-C requires 20-30 minutes of overhead, or 4-5 minutes per point. As noted above, this is a viable method for saving computational overhead when functions evaluations are not especially slow, even if points must be deployed sequentially. This fact further emphasizes the computational benefits of 2-OPT-C over the method of [9].

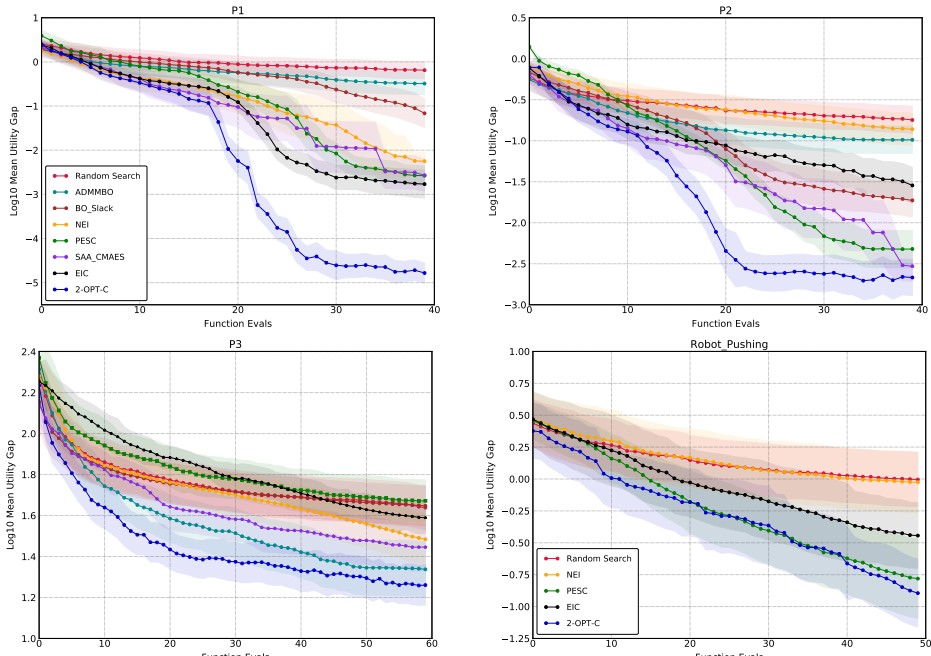

Figure 3: Log10 mean utility gap of Random Search, ADMMBO, BO_Slack, NEI, EIC, PESC, SAA_CMAES, and 2-OPT-C with 95% confidence intervals for P1 (top left), P2 (top right), and P3 (bottom left). Log10 mean utility gap of Random Search, NEI, EIC, PESC, and 2-OPT-C with 95% confidence intervals for robot pushing (bottom right).

### B.3   Experiment Results for the Mean Utility Gap

We report the *mean* utility gap rather than the median, using the same experiments (except for the portfolio optimization problem) reported in §5.2. As in §5.2, we continue to assume sequential evaluations and plot versus the number of function evaluations. Figure 3 provides these additional results. Results for this alternative outcome, the mean, are very similar to those for the median.

### B.4   Additional Non-Myopic Results

We provide the detailed experiment results of §5.3 in Table 1[1]. All the statistics are obtained from Table 1 of [9] and we add results from 2-OPT-C to their table.

---

[1]The performance of EIC as reported in [9] differs from both our results and those in [8]. Since the [9] EIC implementation leveraged complex rollout code, which may have special characteristics, we exclude those results.

| PROBLEM | N | SLSQP | MMA | COBYLA | ISRES | PESC | PM | R-1 | R-2 | R-3 | 2-OPTC |
|---------|---|-------|-----|--------|-------|------|-----|-----|-----|-----|--------|
| P1 | 40 | 0.59 | 0.59 | -0.05 | -0.19 | -2.68 | 0.30 | -4.59 | -4.52 | -4.42 | **-4.92** |
| P2 | 40 | -0.40 | -0.40 | -0.82 | -0.70 | -2.43 | -1.76 | -2.99 | -2.99 | -2.99 | **-3.08** |
| P3 | 60 | 2.15 | 3.06 | 3.06 | 1.68 | 1.66 | 1.79 | 1.48 | 1.31 | 1.35 | **1.28** |

Table 1: Log10 median utility gap. 2-OPT-C's result is presented in the last column. All other statistics are obtained from [9]. The results of the non-myopic rollout method with different horizons are shown in the three columns before 2-OPT-C.

## C   2-OPT-C with Multiple Constraints

Our approach can also be easily generalized to the case with multiple constraints. Here, we provide the form of the gradient estimator $\Gamma(X_1, \widetilde{X}_1, Y)$ with multiple constraints. We focus on independent constraints for simplicity. Our discussion here can be easily generalized to the dependent case, where the multivariate Gaussian probability distribution over the constraints can be calculated through numerical methods [14].

Suppose we have $M$ constraints ($M$ can be 1). Let $Y = (Y_f, \{Y_{g,m}\}_{1 \leq m \leq M})$, where $Y_{g,m}$ is the $m$-th constraint function evaluated at $X_1$. Then similar to §3, if we use $p(y; X_1)$ to denote the distribution of $Y$, i.e., $Y \sim p(y; X_1)$, then

$$
p(y; X_1) = \mathcal{N}\left(
\begin{bmatrix}
\mu_0(X_1) \\
\mu^c_{0,1}(X_1) \\
\mu^c_{0,2}(X_1) \\
\vdots \\
\mu^c_{0,M}(X_1)
\end{bmatrix},
\begin{bmatrix}
K_0(X_1, X_1) & 0 & 0 & \ldots & 0 \\
0 & K^c_{0,1}(X_1, X_1) & 0 & \ldots & 0 \\
0 & 0 & K^c_{0,2}(X_1, X_1) & \ldots & 0 \\
\vdots & \vdots & \vdots & \ddots & \vdots \\
0 & 0 & \ldots & \ldots & K^c_{0,M}(X_1, X_1)
\end{bmatrix}
\right)
$$

where $\mu^c_{0,m}$ and $K^c_{0,m}$ are the posterior mean and posterior variance of the corresponding Gaussian process modeling the $m$-th constraint function for $m = \{1, 2, \cdots, M\}$ respectively. In other words, $p(y; X_1)$ now is the joint posterior distribution of $Y_f$ and $\{Y_{g,m}\}_{1 \leq m \leq M}$ given the information of the past observations $D$ and their objective $f(D)$ and constraint $g_m(D)$, where $g_m$ is the $m$-th constraint function. Let $p(y; \widetilde{X}_1)$ be the importance sampling distribution defined in a similar way to the main paper but under the multiple constraint setting. Then our gradient estimator has the same form as the one in §4:

$$
\Gamma(X_1, \widetilde{X}_1, y) := \nabla \max_{x_2 \in A(\delta)} \alpha(X_1, x_2, y) L(y; X_1, \widetilde{X}_1)
$$

$$
= \left[\nabla \alpha(X_1, x_2^*, Y)\right] L(y; X_1, \widetilde{X}_1) + \alpha(X_1, x_2^*, Y)\left[\nabla p(y; X_1)\right] / p(y; \widetilde{X}_1)
$$

with $x_2^* \in \arg\max_{x_2 \in A(\delta)} \alpha(X_1, x_2, Y)$. Here, $\alpha(X_1, x_2, y)$ is under the multiple constraint setting, which is explicitly written as:

$$
\alpha(X_1, x_2, y) = f_0^* - f_1^* + \mathrm{EI}(f_1^* - \mu_1(x_2), \sigma_1(x_2)^2) \prod_{m=1}^{M} \mathrm{PF}(\mu^c_{1,m}(x_2), (\sigma^c_{1,m}(x_2))^2)
$$

where $f_0^*$ is the best evaluated point satisfying all the constraints so far, i.e. $f_0^* = \min\{f(x) : x \in D, g_m(x) \leq 0 \text{ for } 1 \leq m \leq M\}$. Similarly, $f_1^*$ and $f_2^*$ are the best evaluated points satisfying all the constraints by the end of the first and second stage respectively. Moreover, $\mu_1(x), \mu^c_{1,m}(x)$ and $\sigma_1(x), \sigma^c_{1,m}(x)$ denote the posterior mean and posterior standard deviation of $f(x)$ and $g(x)$ given D and $X_1$.

## D   2-OPT-C Algorithm

The overall description of the algorithm is provided in §4. Here, we provide implementation details of our algorithm and present pseudo-code for 2-OPT-C. This pseudo-code sets $\widetilde{X}_1$ equal to $X_1$.

To reduce the variance of 2-OPT-C, instead of sampling the fantasy values for $Y_f = \{f(x) : x \in X_1\}$ and $Y_g = \{g(x) : x \in X_1\}$ through classic Monte Carlo (MC) methods, we apply the quasi Monte Carlo technique (QMC)

[15], which uses a low-discrepancy sequence as its sample set. In classic MC, the sampled values tend to cluster even when the sampling distribution is uniform. Consequently, the sampled values may not cover the domain of integration particularly efficiently. However, QMC covers the domain of integration more efficiently and thoroughly. Therefore, QMC provides more robust performance using fewer samples compared to classic MC.

In addition, to reduce the computational burden of 2-OPT-C, we leverage the two time scale optimization technique in [16]. The key idea of this technique is that for the nested optimization problem, the optimal solution (in our case, its $x_2^*$) obtained from the previous iteration of the inner problem should remain within a small neighborhood of a current high quality local optimal solution (or the global solution). In other words, we do not need to solve the inner optimization problem when we calculate every gradient copy for our gradient. Instead, for certain gradient copies, we can use the optimal solution previously obtained and directly calculate the gradient copy. We let $k$ denote the frequency of solving the inner optimization. $k = 2$ means that we solve the inner optimization on every other iteration when calculating the gradient.

Let subscripts $r, t, m$ denote the number of restarts, the number of gradient ascent steps, and the number of samples of the gradient. For simplicity, we drop the subscript representing the stage and use $X$ to denote $X_1$. With this notation we summarize our full algorithm for optimizing 2-OPT-C as follows.

---

**Algorithm 1** Estimating $\nabla$ 2-OPT-C$(X)$ with Two Time Scale Optimization

---

1: Initialize $\nabla$ 2-OPT-C$(X) = 0$
2: Initialize $x_{2,\text{previous}}^* = 0$
3: **for** m = 1:M **do**
4:     Use QMC sampling to get $Y_f$ and $Y_g$
5:     **if** $m \neq k$ **then**
6:         Obtain $x_2^*$ by solving $\arg\max_{x_2} \alpha(X, x_2, y)$
7:     **else**
8:         $x_2^* = x_{2,\text{previous}}^*$
9:     **end if**
10:    Plug $x_2^*$ into $\Gamma(X, X, y)$ and compute the $m$-th gradient estimation: $\nabla \widehat{\text{2-OPT}}\text{-C}(X)_m$.
11: **end for**
12: return the estimated gradient $\nabla \widehat{\text{2-OPT}}\text{-C}(X) = \frac{1}{M} \sum_{m=1}^{M} \nabla \widehat{\text{2-OPT}}\text{-C}(X)_m$

---

---

**Algorithm 2** Optimization of 2-OPT-C

---

1: **for** r = 1:R **do**
2:     Draw an initial point or a batch of points $X_{r,0}$ from the Latin hypercube
3:     **for** t = 0:T-1 **do**
4:         Estimate $\nabla$ 2-OPT-C$(X_{r,t})$ using Algorithm 1 and store the result as $G(X_{r,t})$
5:         Update solution $X_{r,t+1} = (1 - \alpha_t)X_{r,t} + \alpha_t G(X_{r,t})$, where $\alpha_t$ is the step size at time t
6:     **end for**
7:     Estimate 2-OPT-C$(X_{r,T})$ using Monte Carlo simulation and store the estimate as $\widehat{\text{2-OPT}}\text{-C}(X_{r,T})$
8: **end for**
9: return $X_{r^*,T}$ and $\widehat{\text{2-OPT}}\text{-C}(X_{r^*,T})$, where $r^* = \max_r \widehat{\text{2-OPT}}\text{-C}(X_{r,T})$

---

## E   Proof of Theorem 1

This section proves Theorem 1. We first restate the theorem and detail the regularity conditions (Assumption 1) it requires.

**Assumption 1.** *We assume*

1. *The prior on the objective function $f$ is a Gaussian Process $f \sim GP(\mu_f(x), K_f(x, x'))$, and the prior on the constraints $g$ is another Gaussian Process $g \sim GP(\mu_g(x), K_g(x, x'))$. These two Gaussian processes are independent.*

2. *$\mu_f$ and $\mu_g$ is continuously differentiable with $x$.*

3. *$K_f(x, x')$ and $K_g(x, x')$ is continuously differentiable with $x$ and $x'$.*

    *4. Given $n$ different points $X = (x_1, x_2, ..., x_n)$, the matrix $K_f(X, X)$ and $K_g(X, X)$ are of full rank.*

Assumption 1.1 models the objective $f$ and constraints $g$ as two independent Gaussian processes, following [6]. Assumption 1.2 and 1.3 assumes the mean and the kernel functions of the two Gaussian processes are smooth. Assumption 1.4 assumes non-degeneracy of the kernel matrix, i.e., the posterior variance at a point is zero if and only if this point is already sampled. All the assumptions are standard in the Bayesian Optimization literature [17].

We now restate Theorem 1 before providing the proof: under Assumption 1, 2-OPT-C$(X_1)$'s partial derivatives exist almost everywhere for any $\delta > 0$. When 2-OPT-C$(X_1)$ is differentiable,

$$\nabla \text{2-OPT-C}(X_1) = \int \Gamma(X_1, \widetilde{X}_1, y) p(y; \widetilde{X}_1) dy.$$

To support the proof, we first define some notation. First, we use $A_\delta^c := A \backslash A_c$ to denote the complementary set of $A_\delta$; clearly, $A_\delta^c$ is an open set. Second, we use $d(X_1, \widetilde{X}_1) := \max_{1 \leq i \leq q} |x^{(i)} - \tilde{x}^{(i)}|$ where $X_1 = (x^{(1)}, x^{(2)}, ..., x^{(q)})$ and $\widetilde{X}_1 = (\tilde{x}^{(1)}, \tilde{x}^{(2)}, ..., \tilde{x}^{(q)})$ are two sets of batched points. $d(X_1, \widetilde{X}_1)$ measure the similarity of $X_1$ and $\widetilde{X}_1$, and $d(X_1, \widetilde{X}_1) = 0$ if and only if $X_1 = \widetilde{X}_1$. Finally, we define

$$\text{2-OPT-C}_\delta(X_1, \widetilde{X}, y) := \max_{x_2 \in A_\delta} \alpha(X_1, x_2, y) L(y; X_1, \widetilde{X}).$$

With this new notation,

$$\text{2-OPT-C}(X_1) = \int \text{2-OPT-C}_\delta(X_1, \widetilde{X}, y) p(y; \widetilde{X}) dy.$$

Before diving into the proof of Theorem 1, we give an overview of all the lemmas needed and their roles in the proof. Lemma 1 bounds the posterior mean and posterior standard deviation of the objective and the constraints functions, which serves a cornerstone to bound $\nabla_{X_1} \text{2-OPT-C}_\delta(X_1, \widetilde{X}, y)$ (Lemma 2). Lemma 3 shows 2-OPT-C$_\delta(X_1, \widetilde{X}, y)$ is absolutely continuous with respect to $X_1$, thus combining Lemma 2 and Lemma 3 shows $\nabla_{X_1} \text{2-OPT-C}_\delta(X_1, \widetilde{X}, y)$ is joint integratable with respect to $X_1 \times y$. Applying Fubini Theorem to interchange the integration of $X_1$ and $y$ proves Theorem 1.

**Lemma 1.** *Given the set of sampled points $D$, batched points $\widetilde{X}_1$ and $\delta' < \delta$, there exists constant $C_0, C_1$ and $C_2$, s.t. for any $y$, $x_2 \in A_\delta$ and $X_1$ s.t. $d(X_1, \widetilde{X}_1) \leq \delta'$, the posterior mean and standard deviation of the objective $f$ and constraints $g$ at $x_2$ satisfy:*

- $|\mu_1(x_2)| \leq C_1 + C_2|y|$, $|\mu_1^c(x_2)| \leq C_1 + C_2|y|$;

- $|\nabla_{X_1} \mu_1(x_2)| \leq C_1 + C_2|y|$, $|\nabla_{X_1} \mu_1^c(x_2)| \leq C_1 + C_2|y|$;

- $C_0 \leq |\sigma_1(x_2)| \leq C_1$, $C_0 \leq |\sigma_1^c(x_2)| \leq C_1$;

- $|\nabla_{X_1} \sigma_1(x_2)| \leq C_1$, $|\nabla_{X_1} \sigma_1^c(x_2)| \leq C_1$.

*When there are multiple constraints, $\sigma_1^c(x_2)$ is understood as the upper triangle matrix by Cholesky decomposition of the covariance matrix at $x_2$.*

*Proof of Lemma 1* Without loss of generality, we assume the set of sampled points is an empty set. Otherwise, we could change the definition of $\mu_f$, $\mu_g$ and $K_f$, $K_g$ to be the posterior mean and kernel function conditioned on the sampled data $D$.

We only prove the inequalities considering the objective function $f$. The proof for inequalities considering the constraints functions $g$ is the same.

Conditioned on objective function $y_f := f(X_1)$ evaluated at $X_1$, the posterior of $f(x_2)$ is a Gaussian distribution with mean $\mu(x_2)$ and variance $\sigma^2(x_2)$, where

$$\mu_1(x_2) := \mu_f(x_2) + K_f(x_2, X_1) K(X_1, X_1)^{-1}(y_f - \mu_f(X_1))$$
$$\sigma_1^2(x_2) := K(x_2, x_2) - K(x_2, X_1) K(X_1, X_1)^{-1} K(X_1, x_2).$$

The bound on the posterior mean $\mu_1(x_2)$ follows from

$$
\begin{aligned}
|\mu_1(x_2)| = |\mu_f(x_2) &+ K_f(x_2, X_1)K(X_1, X_1)^{-1}(y_f - \mu_f(X_1))| \\
\leq &\max_{x_2 \in A_\delta} |\mu_f(x_2)| \\
&+ \max_{x_2 \in A_\delta, d(X_1, X_1')\delta'} |K_f(x_2, X_1)| \max_{d(X_1, X_1')\delta'} |K(X_1, X_1)^{-1}||y_f| \\
&+ \max_{x_2 \in A_\delta, d(X_1, X_1')\delta'} |K_f(x_2, X_1)| \max_{d(X_1, X_1')\leq\delta'} |K(X_1, X_1)^{-1}| \max_{d(X_1, X_1')\leq\delta'} |\mu_f(X_1)|,
\end{aligned}
$$

where $\max_{x_2 \in A_\delta} |\mu_f(x_2)| < \infty$ by the smoothness of $\mu_f$ and compactness of $A_\delta$. Similarly,

$$
\max_{x_2 \in A_\delta, d(X_1, X_1')\delta'} |K_f(x_2, X_1)| < \infty, \quad \max_{d(X_1, X_1')\delta'} |K(X_1, X_1)^{-1}| < \infty, \quad \max_{d(X_1, X_1')\leq\delta'} |\mu_f(X_1)| < \infty,
$$

follows from the compactness of the region $\{X_1 : d(X_1, X_1') \leq \delta'\}$, the smoothness of $K_f$ and $\mu_f$, and the full-rank property of $K_f$.

The bound on $\nabla_{X_1}\mu_1(x_2)$, $\sigma_1(x_2)$ and $\nabla_{X_1}\sigma_1(x_2)$ can be deduced by following the same spirit. □

**Lemma 2.** *Given the set of sampled points $D$, batched points $\widetilde{X}_1$ and $\delta' < \delta$, there exists constant $C_1$ and $C_2$, s.t. for any $y$, $x_2 \in A_\delta$ and $X_1$ s.t. $d(X_1, \widetilde{X}_1) \leq \delta'$,*

- $|\alpha(X_1, x_2, y)| \leq C_1 + C_2|y|$,

- $|\nabla_{X_1}\alpha(X_1, x_2, y)| \leq C_1 + C_2|y|$,

- $|\nabla_{X_1}L(y; X_1, \widetilde{X}_1)| \leq (C_1 + C_2|y|^2)|L(y; X_1, \widetilde{X}_1)|$.

*Proof of Lemma 2:* To prove the first inequality,

$$
\begin{aligned}
|\alpha(X_1, x_2, y)| &= |f_0^* - f_1^* + \mathbb{E}_1[(f_1^* - f(x_2))^+ I(g(x_2) \leq 0)]| \\
&\leq |f_0^*| + |f_1^*| + |\mathbb{E}_1[(f_1^* - f(x_2))^+ I(g(x_2) \leq 0)]| \\
&\leq |f_0^*| + |y| + \mathbb{E}_1[|f_1^* - f(x_2)|] \\
&\leq |f_0^*| + |y| + |f_1^*| + \mathbb{E}_1[|f(x_2)|] \\
&\leq |f_0^*| + |y| + |y| + |\mu_1(x_2)| + |\sigma_1(x_2)| \\
&\leq C_1 + C_2|y|,
\end{aligned}
$$

where applying Lemma 1 to $\mu_1(x_2)$ and $\sigma_1(x_2)$ derives the last inequality.

To prove the second inequality, notice

$$
\alpha(X_1, x_2, Y) = f_0^* - f_1^* + \text{EI}(f_1^* - \mu_1(x_2), \sigma_1(x_2)^2)\text{PF}(\mu_1(x_2)^c, (\sigma_1(x_2)^c)^2),
$$

where $\text{EI}(m, v) := m\Phi(m/\sqrt{v}) + \sqrt{v}\varphi(m/\sqrt{v})$ and $\text{PF}(m^c, v^c) = \Phi\left(-m^c/\sqrt{v^c}\right)$. Notice functions $\Phi, \phi$ along with their derivatives $\Phi', \phi'$ are bounded. Thus, applying the chain rule along with Lemma 1 proves the second inequality.

To prove the third inequality, notice

$$
L(y; X_1, \widetilde{X}_1) = \frac{p(y; X_1)}{p(y; \widetilde{X}_1)}
$$

where

$$
p(y; X_1) = \mathcal{N}\left(\begin{bmatrix} \mu_0(X_1) \\ \mu_0^c(X_1) \end{bmatrix}, \begin{bmatrix} \sigma_0^2(X_1) & 0 \\ 0 & (\sigma_0^c(X_1))^2 \end{bmatrix}\right).
$$

By following an approach similar to the second inequality's proof, it is straightforward to prove the third inequality. □

**Lemma 3.** *Given $\widetilde{X}$, suppose $X_{1,1}$ and $X_{1,2}$ satisfy $d(\widetilde{X}_1, X_{1,1}) < \delta$ and $d(\widetilde{X}_1, X_{1,1}) < \delta$. We can construct the path map $\epsilon \to X_1(\epsilon)$, where $\epsilon \in [0, 1]$ and*

$$
X_1(\epsilon) = \epsilon X_{1,1} + (1 - \epsilon)X_{1,2}.
$$

*Then 2-OPT-C$_\delta(X_1(\epsilon), \widetilde{X}_1, y)$ is absolutely continuous with $X_1(\epsilon)$, i.e.*

$$
\text{2-OPT-C}_\delta(X_1(1), \widetilde{X}_1, y) = \text{2-OPT-C}_\delta(X_1(0), \widetilde{X}_1, y) + \int_0^1 \frac{d}{d\epsilon} \text{2-OPT-C}_\delta(X_1(\epsilon), \widetilde{X}_1, y)d\epsilon.
$$

*In particular,*

$$\frac{d}{d\epsilon} \text{2-OPT-C}_\delta(X_1(\epsilon), \widetilde{X}_1, y)\Big|_{\epsilon=\epsilon'} = \frac{d}{d\epsilon}\left[\alpha(X_1(\epsilon), x_2(\epsilon'), y)L(y; X_1(\epsilon), \widetilde{X}_1)\right]\Big|_{\epsilon=\epsilon'}$$

*where* $x_2(\epsilon') = \arg\max_{x_2} \alpha(X_1(\epsilon'), x_2, y)L(y; X_1(\epsilon'), \widetilde{X}_1)$.

*Proof of Lemma 3* According to Corollary 4 in [18], we only need to check the following three conditions:

- $A_\delta$ is non-empty compact set.
- $\alpha(X_1(\epsilon), x_2, y)L(y; X_1(\epsilon), \widetilde{X}_1)$ is continuous in $x_2$.
- $\frac{d}{d\epsilon}\left[\alpha(X_1(\epsilon), x_2, y)L(y; X_1(\epsilon), \widetilde{X}_1)\right]$ is continuous in $\epsilon$ and $x_2$.

The first condition holds true according to the definition of $A_\delta$. Because the posterior mean and variance is a continuously differentiable function of $X_1$, the second and the third condition can be verified with the chain rule. □

**Lemma 4.** *Given* $\widetilde{X}$, *suppose* $X_{1,1}$ *and* $X_{1,2}$ *satisfy* $d(\widetilde{X}_1, X_{1,1}) < \delta$ *and* $d(\widetilde{X}_1, X_{1,1}) < \delta$. *We can construct the path map* $\epsilon \to X_1(\epsilon)$, *where* $\epsilon \in [0,1]$ *and*

$$X_1(\epsilon) = \epsilon X_{1,1} + (1 - \epsilon)X_{1,2}.$$

*Then*

$$\frac{d}{d\epsilon} \text{2-OPT-C}_\delta(X_1(\epsilon), \widetilde{X}_1, y)$$

*is integrable in the product region* $[0,1] \times \text{Range}(y)$ *with the product measure, where the measure on* $[0,1]$ *is given by the Lebesgue measure and the measure on* $\text{Range}(y)$ *is given by the Gaussian density function*

$$p(y; \widetilde{X}_1) = \mathcal{N}\left(\begin{bmatrix} \mu_0(\widetilde{X}_1) \\ \mu_0^c(\widetilde{X}_1) \end{bmatrix}, \begin{bmatrix} \sigma_0^2(\widetilde{X}_1) & 0 \\ 0 & (\sigma_0^c(\widetilde{X}_1))^2 \end{bmatrix}\right).$$

*Proof of Lemma 4:* According to Lemma 3,

$$\frac{d}{d\epsilon} \text{2-OPT-C}_\delta(X_1(\epsilon), \widetilde{X}_1, y)\big|_{\epsilon=\epsilon'} = \frac{d}{d\epsilon}\left[\alpha(X_1(\epsilon), x_2(\epsilon'), y)L(y; X_1(\epsilon), \widetilde{X}_1)\right]\Big|_{\epsilon=\epsilon'}$$

$$= [\frac{d}{d\epsilon}\alpha(X_1(\epsilon), x_2(\epsilon'), y)]L(y; X_1(\epsilon), \widetilde{X}_1) + \alpha(X_1(\epsilon), x_2(\epsilon'), y)\frac{d}{d\epsilon}L(y; X_1(\epsilon), \widetilde{X}_1)$$

According to Lemma 2, there exists $C_1$ and $C_2$, s.t.

$$|\frac{d}{d\epsilon}\alpha(X_1(\epsilon), x_2(\epsilon'), y)| \leq C_1 + C_2|y|,$$

$$\alpha(X_1(\epsilon), x_2(\epsilon'), y) \leq C_1 + C_2|y|,$$

$$\frac{d}{d\epsilon}L(y; X_1(\epsilon), \widetilde{X}_1) \leq (C_1 + C_2|y|^2)L(y; X_1(\epsilon), \widetilde{X}_1).$$

So, there exists constant $C_1$ and $C_2$, s.t.

$$\frac{d}{d\epsilon} \text{2-OPT-C}_\delta(X_1(\epsilon), \widetilde{X}_1, y) \leq (C_1 + C_2|y|^3)L(y; X_1(\epsilon), \widetilde{X}_1).$$

Thus, Lemma 4 is proved. □

Now we are ready to prove Theorem 1.

*Proof of Theorem 1* Without loss of generality, we assume $d(X_1, \widetilde{X}_1) \leq \frac{\delta}{2}$. Other cases can be proved in a straightforward fashion following our approach here.

When calculating a partial derivative at $X_1$, we choose a point $X_1'$, s.t. $d(X_1', \widetilde{X}_1) \leq \frac{\delta}{2}$ and $X_1' - X_1$ gives the direction in which we are interested. Construct the path map $X_1(\epsilon) = \epsilon X_1 + (1 - \epsilon)X_1'$ where $\epsilon \in [0,1]$.

According to Lemma 3,

$$\text{2-OPT-C}_\delta(X_1(\epsilon), \widetilde{X}_1, y) = \text{2-OPT-C}_\delta(X_1(0), \widetilde{X}_1, y) + \int_0^\epsilon \frac{d}{d\epsilon} \text{2-OPT-C}_\delta(X_1(\epsilon), \widetilde{X}_1, y) d\epsilon.$$

Thus,

$$\begin{aligned}
\text{2-OPT-C}(X_1(\epsilon)) &= \int \text{2-OPT-C}_\delta(X_1(\epsilon), \widetilde{X}_1, y) p(y; \widetilde{X}_1) dy \\
&= \int \text{2-OPT-C}_\delta(X_1(0), \widetilde{X}_1, y) p(y; \widetilde{X}_1) dy + \int p(y; \widetilde{X}_1) dy \int_0^\epsilon \frac{d}{d\epsilon} \text{2-OPT-C}_\delta(X_1(\epsilon), \widetilde{X}_1, y) d\epsilon.
\end{aligned}$$

According to Lemma 4, $\frac{d}{d\epsilon}$ 2-OPT-C$_\delta(X_1(\epsilon), \widetilde{X}_1, y)$ is integratable with respect to $\epsilon \times y$. Thus, by Fubini Theorem,

$$\begin{aligned}
\text{2-OPT-C}(X_1(\epsilon)) &= \int \text{2-OPT-C}_\delta(X_1(0), \widetilde{X}_1, y) p(y; \widetilde{X}_1) dy + \int p(y; \widetilde{X}_1) dy \int_0^\epsilon \frac{d}{d\epsilon} \text{2-OPT-C}_\delta(X_1(\epsilon), \widetilde{X}_1, y) d\epsilon \\
&= \int \text{2-OPT-C}_\delta(X_1(0), \widetilde{X}_1, y) p(y; \widetilde{X}_1) dy + \int_0^\epsilon d\epsilon \int \frac{d}{d\epsilon} \text{2-OPT-C}_\delta(X_1(\epsilon), \widetilde{X}_1, y) p(y; \widetilde{X}_1) dy.
\end{aligned}$$

Thus,

$$\frac{d}{d\epsilon} \text{2-OPT-C}(X_1(\epsilon)) = \int \frac{d}{d\epsilon} \text{2-OPT-C}_\delta(X_1(\epsilon), \widetilde{X}_1, y) p(y; \widetilde{X}_1) dy.$$

When 2-OPT-C$(X_1)$ is differentiable with respect to $X_1$,

$$\begin{aligned}
\nabla \text{2-OPT-C}(X_1) &= \int \nabla \text{2-OPT-C}_\delta(X_1, \widetilde{X}_1, y) p(y; \widetilde{X}_1) dy \\
&= \int \Gamma(X_1, \widetilde{X}_1, y) p(y; X_1) dy.
\end{aligned}$$

## F   Constrained Multi-points Expected Improvement

Here, we formally derive the form for the constrained multi-points expected improvement defined in §2.3, which allows multiple points to be evaluated at each iteration. This idea is first discussed in [19] but not formally defined.

Let $\mathbf{X} = \{x^{(n+1)}, x^{(n+2)}, \dots, x^{(n+q)}\}$ be the set of q candidate points that we consider evaluating next. Let $f^*$ be the best evaluated point subject to the constraint. Then the improvement provided by those points subject to the constraint, $I(\mathbf{X})$, can be written as:

$$\begin{aligned}
I(\mathbf{X}) &= f^* - \min\{f^*, \min\{f(x) : x \in \mathbf{X}, g(x) \le 0\}\} & (3) \\
&= \max\{0, f^* - \min\{f(x) : x \in \mathbf{X}, g(x) \le 0\}\} & (4) \\
&= \max\{0, \max\{f^* - f(x) : x \in \mathbf{X}, g(x) \le 0\}\} & (5) \\
&= [\max\{f^* - f(x) : x \in \mathbf{X}, g(x) \le 0\}]^+ & (6) \\
&= \max_{x \in \mathbf{X}} (f^* - f(x))^+ \cdot \mathbb{1}\{g(x) \le 0\} & (7)
\end{aligned}$$

where $\mathbb{1}\{A\}$ is the indicator function that equals 1 if $A$ is true and 0 elsewhere. The last equality holds true for the following reason. If for any $x \in \mathbf{X}$, $g(x) > 0$, the inner maximization in (4) will be taken over an empty set. In other words, since any point in $\mathbf{X}$ does not satisfy the constraint, there should be no improvement in $\mathbf{X}$. Therefore, the result in (4) should be 0, which is the same as that in (5). In addition, consider the case where there exists at least one $x \in X_1$ such that $g(x) \le 0$ and $f^* > f(x)$, then:

$$\begin{aligned}
I(\mathbf{X}) &= [\max\{f^* - f(x) : x \in \mathbf{X}, g(x) \le 0\}]^+ \\
&= \max\{f^* - f(x) : x \in \mathbf{X}, g(x) \le 0\} \\
&= \max_{x \in \mathbf{X}} (f^* - f(x)) \cdot \mathbb{1}\{g(x) \le 0\} \\
&= \max_{x \in \mathbf{X}} (f^* - f(x))^+ \cdot \mathbb{1}\{g(x) \le 0\}
\end{aligned}$$

Thus, the last equality holds true and we call $I(\mathbf{X}) = \max_{x \in \mathbf{X}} \ (f^* - f(x))^+ \cdot \mathbb{1}\{g(x) \le 0\}$ the *constrained multi-points improvement*.

Then we define the *constrained multi-points expected improvement* $\mathrm{EIC}(\mathbf{X})$ as the expectation of the constrained multi-points improvement:

$$\mathrm{EIC}(\mathbf{X}) := \mathbb{E}\left[I(\mathbf{X})\right] = \mathbb{E}\left[\max_{x \in \mathbf{X}} \ (f^* - f(x))^+ \cdot \mathbb{1}\{g(x) \le 0\}\right]$$

where $\mathbb{E}[\cdot]$ is the expectation taken with respect to the posterior distribution given the current observations $D$ and their objective values $f(D)$ and constraint values $g(D)$.