# OpenReview forum: "Constrained Two-step Look-Ahead Bayesian Optimization"
_NeurIPS.cc/2021/Conference — NeurIPS 2021 Poster_

### Official Review · Reviewer_HPhn · 2021-07-13

**Rating:** 7
**Confidence:** 4

**Summary:**

This paper deals with non-myopic BO under inequality constraints. In particular, recent results on the efficient approximation and optimization of the two-step expected improvement criterion are revisited in the considered constrained case, leading to an efficient gradient estimation scheme relying both on importance sampling and on a clever application of the envelope theorem. Experimental results confirm that the developed method with the proposed importance distribution perform better than baseline approaches on the considered examples.

**Limitations And Societal Impact:**

Ok

**Main Review:**

This is very neat work. While I think it deserves to be published as it adequately addresses an important class of BO problems, my main reservation would be that it is relatively incremental with respect to „Practical two-step lookahead Bayesian optimization“ (Wu and Frazier, Neurips 2019). Indeed both key ingredients of a) the clever use of the envelope theorem, and b) the use of importance sampling in a related context of acquisition function evaluation/differentiation are already presented there, yet in an unconstrained setting. Considering this, the current paper, while being excellent in several regards, somehow feels like a nice extension of the former paper. Spending less room on the basics (that can be found in the 2019 paper or that amount to routine calculations to be placed for instance in appendix) and more on the actual novelties (that could be further developed, e.g. choices regarding importance sampling) could help making this nice paper an outstanding one. Also, relations to some more developments in this very active field of research could be further explored, for instance with the approach of „Efficient Nonmyopic Bayesian Optimization via One-Shot Multi-Step Trees“ (Jiang et al., Neurips 2020) and the strategies of „Efficient Rollout Strategies for Bayesian Optimization“ (Lee et al.. UAI 2020).

Post response from the authors: increased my score by one unit. Congrats on the convincing responses and promising test case results.


**Time Spent Reviewing:**

2.5

---

> ### Author Response · Authors · 2021-08-11
> **Author response to reviewer HPhn**
>
>
> We would like to thank you for your feedback and questions. We are glad that you found our work “very neat”, “being excellent in several regards” and that you feel it “deserves to be published”. Below, we respond to the questions and comments raised in this review.
>
> $\bf{Q1:}$ This is very neat work. While I think it deserves to be published as it adequately addresses an important class of BO problems, my main reservation would be that it is relatively incremental with respect to “Practical two-step lookahead Bayesian optimization” (Wu and Frazier, Neurips 2019). Indeed both key ingredients of a) the clever use of the envelope theorem, and b) the use of importance sampling in a related context of acquisition function evaluation/differentiation are already presented there, yet in an unconstrained setting. Considering this, the current paper, while being excellent in several regards, somehow feels like a nice extension of the former paper. Spending less room on the basics (that can be found in the 2019 paper or that amount to routine calculations to be placed for instance in appendix) and more on the actual novelties (that could be further developed, e.g. choices regarding importance sampling) could help making this nice paper an outstanding one.
>
> $\bf{A1:}$ We are grateful that the reviewer appreciated the work.
> A significant source of novelty in our paper relative to Wu & Frazier (2019) is our likelihood ratio estimator of the gradient of the acquisition function. Wu & Frazier (2019) uses infinitesimal perturbation analysis (IPA) in the unconstrained setting. That method works well there, but fails in the constrained setting, as we explain in lines 75-77 and 196-202. This requires us to abandon their approach to estimating the gradient of the acquisition function, instead of developing our new likelihood ratio method. We believe that this methodology may also be applicable more broadly in Bayesian optimization to other settings whose acquisition function is an expectation over Monte Carlo samples of discontinuous functions.
>
> Another source of novelty relative to Wu & Frazier (2019) is our realization that being non-myopic is even more important in the constrained setting than in the unconstrained setting. Our numerical experiments show a much more significant improvement over myopic methods than those in Wu & Frazier (2019) and we believe that this is for the reason articulated in lines 166-182.
>
> We will emphasize this novelty in the revision. While shortening some material in Section 3 that extends Wu & Frazier (2019) would allow us to more quickly reach the material that uses a new and different approach in Sections 4 and 5, other reviewers have emphasized the importance of having the paper be self-contained, and so we must strike a balance.
>
> $\bf{Q2:}$ Also, relations to some more developments in this very active field of research could be further explored, for instance with the approach of “Efficient Nonmyopic Bayesian Optimization via One-Shot Multi-Step Trees” (Jiang et al., Neurips 2020) and the strategies of “Efficient Rollout Strategies for Bayesian Optimization” (Lee et al.. UAI 2020).
>
> $\bf{A2:}$
> Thank you for these suggestions. We agree that there are many exciting opportunities suggested by the connections between our work and these other papers or non-myopic BayesOpt. We will discuss this in the paper as potential future work, especially opportunities for leveraging our likelihood-ratio-based approach as a way to overcome discontinuities in acquisition functions within the larger frameworks for looking more than two steps ahead in Jiang et al. and Lee at al. It would also be interesting to roll out 2-OPT-C within the approach of Lee at al. as a way to deliver greater query efficiency than rolling out EIC.

---

### Official Review · Reviewer_Strq · 2021-07-16

**Rating:** 7
**Confidence:** 3

**Summary:**


This paper introduces a two-step look-ahead expected improvement policy for Bayesian optimization with inequality constraints. The authors argue that infinitesimal perturbation analysis and sample average approximation methods create discontinuities during the acquisition function maximization of two-step expected improvement for BO with inequalities. The authors propose a solution based on a likelihood ratio approach that computes an unbiased estimator for the gradient of two-step expected improvement with constraints (Theorem 1). Empirical results show that the proposed method is effective and outperforms baselines.

**Ethical Concerns:**

N/A.

**Limitations And Societal Impact:**

Yes.

**Main Review:**

**Strengths**

Bayesian optimization with constraints is an important topic for the NeurIPS community.  The authors justify the importance of the paper by describing the limitation of the standard approach (Figure 2). The construction of the unbiases estimator is a little bit hard to follow, but I think I understood the overall idea.

**Weakness**

This paper would be more substantial if the authors tested their method on more practical functions. Experiments were conducted over only three functions. Why not test on the Reacting flow problem (P4) too [1]?

**After rebuttal**
Thanks for the response. I think adding this example to the paper would make it considerably strongger.

**Time Spent Reviewing:**

3

---

> ### Author Response · Authors · 2021-08-11
> **Author response to reviewer Strq featuring real world experiment**
>
> We would like to thank you for your feedback and questions. We are glad that you found the problem we study “an important topic” and that we “justify the importance of the paper by describing the limitation of the standard approach”. Below, we respond to the questions and comments raised in this review.
>
> $\bf{Q1:}$ This paper would be more substantial if the authors tested their method on more practical functions. Experiments were conducted over only three functions. Why not test on the Reacting flow problem (P4) too [1]?
>
> $\bf{A1:}$ By the time of submitting this paper, we couldn’t set up the experiment according to the description of the Reacting Flow problem (P4) in [1]. However, during the response period, we ran experiments on a real-world portfolio optimization problem, described above in detail in response to reviewer 8KpE.  We reproduce that discussion here for convenience:
>
> Since reading these reviews, we have completed initial experiments studying a real-world portfolio optimization problem based on Boyd (2017), as studied in a BO context by Astudillo & Frazier (2020). In this problem, we optimize an algorithmic trading strategy to maximize a portfolio’s mean annualized return while constraining the portfolio’s risk, as measured by the standard deviation of the return. The portfolio simulation uses CVXPortfolio and real-world market data, as described in Boyd (2017). A single function evaluation using CVXPortfolio takes between 10 and 15 minutes. 50 replications have been completed of both 2-OPT-C, EIC and PESC, using 30 iterations and 3 initial points. At least one of the initial points is feasible.
> We find that 2-OPT-C provides a significant improvement. At iteration 0, the mean annualized return of the best feasible trading strategy for each of the three methods is near 5.1%. At an early stage (iteration 5), the three methods have similar performance with 2-OPT-C at  6.02%, EIC at 5.95%, PESC at 5.79%. Afterwards, 2-OPT-C begins to outperform the other two methods by a larger margin as the optimization proceeds. At iteration 15, 2-OPT-C achieves a value of 6.83% compared to the value 6.44% achieved by EIC and 6.28% by PESC. At the end (iteration 30), the best feasible value found by 2-OPT-C is 7.08% compared to 6.60% by EIC and 6.54% by PESC.
> This improvement provided by 2-OPT-C is significant and practically meaningful. Improving annualized mean return by 0.48% on a financial portfolio would be viewed as a significant addition of value by a hedge fund, investment bank or financial advisor. On a base investment of 1M compounded over 10 years, this would result in an additional 87K in investment income. While a competing method (EIC or PESC) might find a portfolio with a return as good as the one found by OPT-C if allowed more evaluations of the objective and constraint, this might not be possible when interacting with a client asking for a custom portfolio design as evaluating the objective and constraint 30 times already consumes roughly 6 hours of computation.
> We will include this experiment in the final version of the paper.
> We also plan to run experiments on at least one additional real-world problem, such as the reacting flow problem from [1], the hyperparameter tuning of MLP on MNIST data, or the robot pushing problem described in Z. Wang and S. Jegelka (2017).
>
> [1] Remi Lam and Karen Willcox. Lookahead bayesian optimization with inequality constraints. In I. Guyon, U. V. Luxburg, S. Bengio, H. Wallach, R. Fergus, S. Vishwanathan, and R. Garnett, editors, Advances in Neural Information Processing Systems 30, pages 1890–1900. Curran Associates, Inc., 2017.

---

> ### Author Response · Authors · 2021-09-02
> **Post rebuttal response to reviewer Strq with additional experiment results**
>
> We are glad that our previous responses clarified some of the reviewer’s concerns and would like to thank the reviewer for raising the rating of our paper. During the post rebuttal period, we also have run additional experiments on the three synthetic functions and also a new real-world experiment (robot pushing problem) besides the portfolio optimization problem. We reproduce the new results here for convenience.
>
> **New Experiments:**
>
> We conducted additional experiments during the rebuttal period to address the reviewers’ comments. We added four additional benchmark methods: ADMMBO [Ariafar et al., 2019], NEI [Letham et al., 2019], ALBO [Picheny etal., 2016] and random search (RS). We also added a more realistic problem, based on the robot pushing problem introduced in [Wang and Jegelka, 2017].  In addition, we added sample average approximation implemented using CMAES (SAA_CMAES) as a benchmark for problems P1 and P2. The result for P3 can be found in Figure 3 in the main paper. 2-OPT-C substantially outperforms benchmarks in all of these problems. Here we give details of these new experiments.
>
> ***New Benchmarks for Synthetic functions:***
>
> On synthetic problems, i.e., P1, P2, and P3 in the paper, we added the four additional methods mentioned above for comparison. We use the same experimental setup for these three problems as described in the Appendix. In detail, for each algorithm, we ran 150 replications on each problem with three initial points. We ran 40, 40, and 60 iterations for P1, P2, and P3, respectively, for each replication. The results are provided below.
>
> For P1, the (log10) median regret of each of the benchmarked algorithms starts between 0.3 and 0.4. After 20 iterations, RS, ADMMBO, ALBO, NEI, and SAA_CMAES achieve a median regret of 0, -0.18, -0.2, -2, and -2.3 respectively, while 2-OPT-C achieves a substantially smaller median regret of -3.6. After 40 iterations, the median regret of RS, ADMMBO, ALBO, NEI, and SAA_CMAES are -0.2, -0.5, -1.3, -2.2, and -2.9 respectively. The median regret of 2-OPT-C is substantially lower, at -4.8.
>
> | P1 (metric: log_10 median regret) | RS| ADMMBO|ALBO|NEI|SAA_CMAES|2-OPT-C|
> |:---|:----:|:---:|:---:|:---:|:---:|:---:|
> |Iteration 20| 0|-0.18|-0.2|-2.0|-2.3|-3.6|
> |Iteration 40|-0.2|-0.5|-1.3|-2.2|-2.9|-4.8|
>
> For P2, the (log10) median regret of each of the benchmarked algorithms starts between -0.1 and -0.15. After 20 iterations, RS, ADMMBO, ALBO, NEI, and SAA_CMAES achieve a median utility of -0.6, -0.9, -1.25, -0.61, and -2.25 respectively, while 2-OPT-C achieves a much smaller median regret of -3. After 40 iterations, the median utility of RS, ADMMBO, ALBO, NEI, and SAA_CMAES are -0.75, -1.1, -1.8, -0.95, and -2.8 respectively. The median utility of 2-OPT-C is much smaller, at -3.2.
>
> | P2 (metric: log_10 median regret) | RS|ADMMBO|ALBO|NEI|SAA_CMAES|2-OPT-C|
> |:---|:----:|:---:|:---:|:---:|:---:|:---:|
> |Iteration 20| -0.6|-0.9|-1.25|-0.61|-2.25|-3.0|
> |Iteration 40| -0.75|-1.1|-1.8|-0.95|-2.8|-3.2|
>
> For P3, the (log10) median regret of each of the benchmarked algorithms starts between 2.2 and 2.4. After 30 iterations, RS, NEI, and ALBO achieve similar median regret around 1.75, while ADMMBO is able to achieve a median regret of 1.48 and 2-OPT-C achieves an even smaller median regret of 1.35. After 60 iterations, the median regret of RS and ALBO is 1.67. The median regret of NEI is 1.51. Both ADMMBO and 2-OPT-C achieve the smallest median regret, 1.27.
>
> |P3 (metric: log_10 median regret)|RS|ADMMBO|ALBO|NEI|2-OPT-C|
> |:---|:----:|:---:|:---:|:---:|:---:|
> |Iteration 30| 1.75|1.48|1.75|1.75|1.35|
> |Iteration 60| 1.67|1.27|1.67|1.51|1.27|
>
> In the experiments P1, P2, and P3, we notice that non-myopic policies (2-OPT-C and SAA_CMAES) generally perform better than myopic policies (ADMMBO, NEI, ALBO, EIC, and PESC). However, due to the discontinuity introduced by SAA, while SAA_CMAES outperforms the myopic methods in P2 and P3, it underperforms 2-OPT-C and also underperforms EIC in P1.
>
> ***Portfolio Optimization Problem.***
>
> This experiment studies a real-world portfolio optimization problem based on Boyd (2017), as studied in a BO context by Astudillo & Frazier (2020). In this problem, we optimize an algorithmic trading strategy to maximize a portfolio’s mean annualized return while constraining the portfolio’s risk, as measured by the standard deviation of the return. The portfolio simulation uses CVXPortfolio and real-world market data, as described in Boyd (2017). A single function evaluation using CVXPortfolio takes between 10 and 15 minutes. 50 replications have been completed for both 2-OPT-C, EIC and PESC, using 30 iterations and 3 initial points. At least one of the initial points is feasible.
> We find that 2-OPT-C provides a significant improvement.
>
> At iteration 0, the mean annualized return of the best feasible trading strategy for each of the three methods is near 5.1%. At an early stage (iteration 5), 2-OPT-C and EIC have better performances with values at  6.02% and 5.95% respectively compared to PESC at 5.79% and NEI at 5.62%. Afterwards, 2-OPT-C begins to outperform the other three methods by a larger margin as the optimization proceeds. At iteration 15, 2-OPT-C achieves a value of 6.83% compared to the value of 6.44% achieved by EIC, 6.28% by PESC, and 5.97% by NEI. At the end (iteration 30), the best feasible value found by 2-OPT-C is 7.08% compared to 6.60% by EIC, 6.54% by PESC, and 6.29% by NEI.
>
> This improvement provided by 2-OPT-C is significant and practically meaningful. Improving annualized mean return by 0.48% on a financial portfolio would be viewed as a significant addition of value by a hedge fund, investment bank or financial advisor. On a base investment of 1M compounded over 10 years, this would result in an additional 87K in investment income. While a competing method (EIC or PESC) might find a portfolio with a return as good as the one found by OPT-C if allowed more evaluations of the objective and constraint, this might not be possible when interacting with a client asking for a custom portfolio design as evaluating the objective and constraint 30 times already consumes roughly 6 hours of computation.
>
> |Portfolio Optimization (annualized return)|EIC|PESC|NEI|2-OPT-C|
> |:---|:----:|:---:|:---:|---:|
> |Iteration 5|5.95 %|5.79 %|5.62 %|6.02 %|
> |Iteration 15|6.44 %|6.28 %|5.97 %|6.83 %|
> |Iteration 30|6.60 %|6.54 %|6.29 %|7.08 %|
>
> ***Robot Pushing Problem:***
>
> We consider a robot pushing problem based on [Wang and Jegelka, 2017], where the robot pushes an object from the origin, i.e., $L_{\text{object}} = (0,0)$, to an unknown target location $L_{\text{target}} \in [-5,5]^2$. The parameters to be optimized are the location of the robot, i.e., $L_{\text{robot}} = (x_{\text{robot}}, y_{\text{robot}})$ and the duration of the push $t \in [1, 30]$. Therefore, the decision variables to optimize are $\set{x_{\text{robot}}, y_{\text{robot}}, t}$. The objective is to minimize the distance between the location of the object after being pushed and the target location, namely the $L_2$ norm of  $L_{\text{object after push}} -  L_{\text{target}}$. The cost function is the energy used by the robot for pushing the object which is $|| L_{\text{object after push}}|| + \epsilon$, where $\epsilon$ is some noise.
>
> We ran 50 replications for this experiment. For each experiment, we uniformly draw a target location. In other words, we have 50 different target locations uniformly distributed across the replications performed. In each replication, we start with three initial points and run 50 iterations. Here we benchmarked four algorithms NEI, EIC, PESC, and 2-OPT-C. (We also have experiment results for NEI on the portfolio optimization problem and will add that to the final version of the paper. 2-OPT-C outperformed it.) Here, we report the median of the distance between the object after being pushed and the target location of 50 replications.
>
> In the beginning, the median distances for the four algorithms are in the range of 2.5 to 2.6. After 25 iterations, the median distances for NEI and EIC are roughly 1.2 and 0.6, while PESC and 2-OPT-C achieve a median distance of 0.3. After we exhaust the evaluation budget, NEI has a median distance of 0.86 and the median distance for the EIC is 0.35. In comparison, the PESC and 2-OPT-C are able to achieve a smaller median distance of 0.12.
>
> |Robot Pushing|EIC|PESC|NEI|2-OPT-C|
> |:---|:----:|:---:|:---:|---:|
> |Iteration 25|0.6|0.3|1.2|0.3|
> |Iteration 50|0.35|0.12|0.86|0.12|
>
> References:
>
> [Ariafar et al., 2019] Setareh Ariafar, Jaume Coll-Font, Dana Brooks, and Jennifer Dy. Admmbo: Bayesian optimization with unknown constraints using admm. Journal of Machine Learning Research, 20(123):1–26, 2019.
>
> [Letham et al., 2019] Benjamin Letham, Brian Karrer, Guilherme Ottoni, Eytan Bakshy, et al. Constrained bayesian optimization with noisy experiments. Bayesian Analysis, 14(2):495–519, 2019.
>
> [Picheny et al., 2016] Victor Picheny, Robert B Gramacy, Stefan Wild, and Sebastien Le Digabel. Bayesian op- timization under mixed constraints with a slack-variable augmented lagrangian. In D. Lee, M. Sugiyama, U. Luxburg, I. Guyon, and R. Garnett, editors, Advances in Neural Information Processing Systems, volume 29, pages 1435–1443. Curran Associates, Inc., 2016.
>
> [Wang and Jegelka, 2017] Zi Wang and Stefanie Jegelka. Max-value entropy search for efficient Bayesian optimization. In International Conference on Machine Learning, pp. 3627–3635, 2017

---

### Official Review · Reviewer_uDRU · 2021-07-17

**Rating:** 6
**Confidence:** 3

**Summary:**

The paper proposes a method for nonmyopic BO with inequality constraints. The method leverages on 2-step expected improvement with constraints and utilizes reparameterization trick to optimize the acquisition function.

**Ethical Concerns:**

-

**Limitations And Societal Impact:**

-

**Main Review:**

The clarity and the structure of the paper could be greatly improved. The problem setting is not well defined which makes assessing the contributions of the paper challenging. The key concepts and methods are either not not formally introduced (nonmyopic problems,IPA and SAA methods),  introduced in different papers (problem settings for experiments) or introduced in Appendix (Assumption 1 in Theorem 1). Therefore, I believe the paper requires significant re-writing and cannot be accepted in the current form.

A few questions to the authors:

line 138. what kernel is used in the problem?

Line 147. x_2 is a single point or a batch of points? The first stage selects a batch of q points, but the second one seems to select a single point.

Line 149 - best with respect to what?

Line 152 shouldn’t f_1* be a random variable? Then how it is moved out of expectation?

The acquisition function 2-OPT-C doesn’t involve constraint function g at all, so how are the constraints evaluated?

Line 169. What does “tight” mean in this context?

Line 174 - how is the value of 50% obtained?

Line 190 Am I right  that the authors consider noise-free case of BO?

Line 255. Putting Assumption 1 to appendix makes the submission not self-sufficient.

Line 269. The experimental settings are not well-defined.

Line 277. What is the utility gap?

Line 292. Is such comparison with Rollout fair?


-- Post-rebuttal


I would like to thank the authors for answering my questions in their response. I am quite impressed by the amount of work authors have completed during the rebuttal phase in order to improve their paper, and I will increase my score to 6.

**Time Spent Reviewing:**

2

---

> ### Author Response · Authors · 2021-08-11
> **Author response to reviewer uDRU**
>
> We would like to thank the reviewer for all your feedback and questions. In particular, we appreciate your feedback about the clarity of the paper and see a clear avenue to addressing it in a revision. While the request to spend more time on background material should be balanced with reviewer HPhn’s request to spend less time on this same material, your feedback nonetheless suggests ways to improve clarity without adding length.
>
> We were glad that you did not raise any technical concerns with our submission, nor any concerns about novelty or significance.  Given that a detailed response to each request for clarification is offered below and that there is a clear path toward making the suggested edits to improve clarity, we hope that you would consider raising your score after discussing with others on the reviewing team.
>
> $\bf{Q1:}$ The clarity and the structure of the paper could be greatly improved. The problem setting is not well defined which makes assessing the contributions of the paper challenging. The key concepts and methods are either not not formally introduced (nonmyopic problems, IPA and SAA methods), introduced in different papers (problem settings for experiments) or introduced in Appendix (Assumption 1 in Theorem 1). Therefore, I believe the paper requires significant re-writing and cannot be accepted in the current form.
>
> $\bf{A1:}$ We appreciate your feedback about the clarity of the paper. The problem setting is defined on lines 23-26, IPA in the context of constrained BO is explained on lines 197-202 and SAA in this same context is explained on lines 203-208. While the problem setting of constrained BO and the concept of nonmyopic BO are both widely known, we see opportunities to introduce these concepts more clearly. In addition, while IPA and SAA are known by a subset of the BO community, they are less widely known than constrained and nonmyopic BO.
> In the final version of the paper, we will move Assumption 1 into the main paper and more clearly define concepts before they are referenced, including “non-myopic”, IPA and SAA.
>
>
>
> $\bf{Q2:}$ Line 138. what kernel is used in the problem?
>
> $\bf{A2:}$ Due to the page limit, the experimental details are in the Appendix, Section A. We used ARD square-exponential kernels for both objectives and constraints as mentioned in A.1 of the Appendix.
>
> $\bf{Q3:}$ Line 147, x_2 is a single point or a batch of points? The first stage selects a batch of q points, but the second one seems to select a single point.
>
> $\bf{A3:}$ x_2 is a single point. We use capital letters to denote a batch of points, such as X_1. Using a single point in the second stage allows us to compute the value of the second stage analytically, which makes our method faster. Using a batch in the first stage allows us to recommend batches of points.
>
> $\bf{Q4:}$ Line 149 - best with respect to what?
>
> $\bf{A4:}$ In Line 149, “best” means “lowest objective function value”. Expanding on this, when the text says “Let $f_1^*$ and $f_2^*$ be the best evaluated feasible point by the end of the first and second stage, respectively” this means $f_1^*$ is the minimum of f_0^* and the objective function values of all feasible points we observe in stage 1, i.e.,
> $$f_1^* = \min (f_0^*, \min_{x \in X_1, g(x) <= 0} f(x) )$$
>
> $f_2^*$ is defined similarly: $$f_2^* = \min(f_1^*, f(x_2))$$ if $g(x_2) \leq 0$ and $f_2^* = f_1^*$ otherwise.
>
> $\bf{Q5:}$ Line 152 shouldn’t f_1* be a random variable? Then how it is moved out of expectation?
>
> $\bf{A5:}$ Reviewer 8KpE also asked this question. We took a shortcut, skipping a step in moving from the LHS to the RHS in the equation after line 152 and did not realize that this could be confusing. We will revise the paper to include this step. The full explanation for why the LHS and RHS are equal to the RHS in the equation after line 152 is:
> $$E_0[\max_{x2}(f_0^* - f_2^*)]
> = E_0[ E_1[\max_{x2} (f_0^* - f_1^* + f_1^* - f_2^*) | f(X1), g(X1)]]
> = E_0[ f_0^* - f_1^* + \max_{x2} E_1 (f_1^* - f_2^*)] $$
> The second line uses the law of total expectation and also adds and subtracts f_1^*. The third line uses that f_0^* and f_1^* are known constants under the posterior distribution over which E_1 takes the expectation.
>
> $\bf{Q6:}$ The acquisition function 2-OPT-C doesn’t involve constraint function g at all, so how are the constraints evaluated?
>
> $\bf{A6:}$ The acquisition function 2-OPT-C does involve the constraint g. The Gaussian process over g influences the PF term in the expression in line 164. Also, f_1* depends on whether the constraint g is satisfied or violated.
>
> $\bf{Q7:}$ Line 169. What does “tight” mean in this context?
>
> $\bf{A7:}$ “Tight” means that the best feasible point is at or near the boundary between the feasible region (those points where the constraint is satisfied) and the infeasible region (points where the constraint is not satisfied). This terminology is widely used in constrained optimization.
>
>
> $\bf{Q8:}$ Line 174 - how is the value of 50% obtained?
>
> $\bf{A8:}$ This paragraph discusses points close to the boundary between feasibility and infeasibility, where the posterior mean of the Gaussian process on the constraint function (g) is close to 0. (Recall that the constraint is that g(x) must be less than or equal to 0.)
> When the mean of the posterior on g is exactly 0, the probability of feasibility (the probability that g(x) <= 0) under the posterior is 50%. This is because the probability that a mean-0 normally distributed random variable falls below 0 is 50%.
> When the mean on the posterior on g is not exactly 0 but is close, the probability of feasibility is the probability that a normal random variable with a small mean falls below 0. This is close to 50%.
>
> $\bf{Q9:}$ Line 190 Am I right that the authors consider noise-free case of BO?
>
> $\bf{A9:}$ Yes, as we mention in line 26, we consider the noise-free setting. Line 190 rewrites Y_f and Y_g using the reparameterization trick.
>
> $\bf{Q10:}$ Line 255. Putting Assumption 1 to the appendix makes the submission not self-sufficient.
>
> $\bf{A10:}$ We will move it into the main paper.
>
> $\bf{Q11:}$ Line 269. The experimental settings are not well-defined.
>
> $\bf{A11:}$ More details about the experimental setup and settings are found in Section A of the appendix. We are happy to answer specific questions.
>
> $\bf{Q12:}$ Line 277. What is the utility gap?
>
> $\bf{A12:}$ The utility gap is defined in the Section A.1 of the appendix. This is a standard measure of performance in Bayesian optimization.
>
> $\bf{Q13:}$ Line 292. Is such comparison with Rollout fair?
>
> $\bf{A13:}$ Yes. We perform comparison with the rollout method of [1] using the same experimental settings used in that paper, as described in the remainder of the paragraph beginning on line 292. The authors of that paper presumably chose these settings because they felt that they fairly represented the performance of their method.
>
> [1]: Remi Lam and Karen Willcox. Lookahead bayesian optimization with inequality constraints. In I. Guyon, U. V. Luxburg, S. Bengio, H. Wallach, R. Fergus, S. Vishwanathan, and R. Garnett, editors, Advances in Neural Information Processing Systems 30, pages 1890–1900. Curran Associates, Inc., 2017.

---

> ### Author Response · Authors · 2021-09-02
> **Post rebuttal response to reviewer uDRU with additional experiment results**
>
> Since we haven’t received any feedback from the reviewer after the initial round of rebuttal, we hope the reviewer could reconsider the rating given our responses and new experimental results. We reproduce the discussions of the new numerical results besides the portfolio optimization below for the reviewer’s convenience.
>
> **New Experiments:**
>
> We conducted additional experiments during the rebuttal period to address the reviewers’ comments. We added four additional benchmark methods: ADMMBO [Ariafar et al., 2019], NEI [Letham et al., 2019], ALBO [Picheny etal., 2016] and random search (RS). We also added a more realistic problem, based on the robot pushing problem introduced in [Wang and Jegelka, 2017].  In addition, we added sample average approximation implemented using CMAES (SAA_CMAES) as a benchmark for problems P1 and P2. The result for P3 can be found in Figure 3 in the main paper. 2-OPT-C substantially outperforms benchmarks in all of these problems. Here we give details of these new experiments.
>
> ***New Benchmarks for Synthetic functions:***
>
> On synthetic problems, i.e., P1, P2, and P3 in the paper, we added the four additional methods mentioned above for comparison. We use the same experimental setup for these three problems as described in the Appendix. In detail, for each algorithm, we ran 150 replications on each problem with three initial points. We ran 40, 40, and 60 iterations for P1, P2, and P3, respectively, for each replication. The results are provided below.
>
> For P1, the (log10) median regret of each of the benchmarked algorithms starts between 0.3 and 0.4. After 20 iterations, RS, ADMMBO, ALBO, NEI, and SAA_CMAES achieve a median regret of 0, -0.18, -0.2, -2, and -2.3 respectively, while 2-OPT-C achieves a substantially smaller median regret of -3.6. After 40 iterations, the median regret of RS, ADMMBO, ALBO, NEI, and SAA_CMAES are -0.2, -0.5, -1.3, -2.2, and -2.9 respectively. The median regret of 2-OPT-C is substantially lower, at -4.8.
>
> | P1 (metric: log_10 median regret) | RS| ADMMBO|ALBO|NEI|SAA_CMAES|2-OPT-C|
> |:---|:----:|:---:|:---:|:---:|:---:|:---:|
> |Iteration 20| 0|-0.18|-0.2|-2.0|-2.3|-3.6|
> |Iteration 40|-0.2|-0.5|-1.3|-2.2|-2.9|-4.8|
>
> For P2, the (log10) median regret of each of the benchmarked algorithms starts between -0.1 and -0.15. After 20 iterations, RS, ADMMBO, ALBO, NEI, and SAA_CMAES achieve a median utility of -0.6, -0.9, -1.25, -0.61, and -2.25 respectively, while 2-OPT-C achieves a much smaller median regret of -3. After 40 iterations, the median utility of RS, ADMMBO, ALBO, NEI, and SAA_CMAES are -0.75, -1.1, -1.8, -0.95, and -2.8 respectively. The median utility of 2-OPT-C is much smaller, at -3.2.
>
> | P2 (metric: log_10 median regret) | RS|ADMMBO|ALBO|NEI|SAA_CMAES|2-OPT-C|
> |:---|:----:|:---:|:---:|:---:|:---:|:---:|
> |Iteration 20| -0.6|-0.9|-1.25|-0.61|-2.25|-3.0|
> |Iteration 40| -0.75|-1.1|-1.8|-0.95|-2.8|-3.2|
>
> For P3, the (log10) median regret of each of the benchmarked algorithms starts between 2.2 and 2.4. After 30 iterations, RS, NEI, and ALBO achieve similar median regret around 1.75, while ADMMBO is able to achieve a median regret of 1.48 and 2-OPT-C achieves an even smaller median regret of 1.35. After 60 iterations, the median regret of RS and ALBO is 1.67. The median regret of NEI is 1.51. Both ADMMBO and 2-OPT-C achieve the smallest median regret, 1.27.
>
> |P3 (metric: log_10 median regret)|RS|ADMMBO|ALBO|NEI|2-OPT-C|
> |:---|:----:|:---:|:---:|:---:|:---:|
> |Iteration 30| 1.75|1.48|1.75|1.75|1.35|
> |Iteration 60| 1.67|1.27|1.67|1.51|1.27|
>
> In the experiments P1, P2, and P3, we notice that non-myopic policies (2-OPT-C and SAA_CMAES) generally perform better than myopic policies (ADMMBO, NEI, ALBO, EIC, and PESC). However, due to the discontinuity introduced by SAA, while SAA_CMAES outperforms the myopic methods in P2 and P3, it underperforms 2-OPT-C and also underperforms EIC in P1.
>
> ***Portfolio Optimization Problem.***
>
> This experiment studies a real-world portfolio optimization problem based on Boyd (2017), as studied in a BO context by Astudillo & Frazier (2020). In this problem, we optimize an algorithmic trading strategy to maximize a portfolio’s mean annualized return while constraining the portfolio’s risk, as measured by the standard deviation of the return. The portfolio simulation uses CVXPortfolio and real-world market data, as described in Boyd (2017). A single function evaluation using CVXPortfolio takes between 10 and 15 minutes. 50 replications have been completed for both 2-OPT-C, EIC and PESC, using 30 iterations and 3 initial points. At least one of the initial points is feasible.
> We find that 2-OPT-C provides a significant improvement.
>
> At iteration 0, the mean annualized return of the best feasible trading strategy for each of the three methods is near 5.1%. At an early stage (iteration 5), 2-OPT-C and EIC have better performances with values at  6.02% and 5.95% respectively compared to PESC at 5.79% and NEI at 5.62%. Afterwards, 2-OPT-C begins to outperform the other three methods by a larger margin as the optimization proceeds. At iteration 15, 2-OPT-C achieves a value of 6.83% compared to the value of 6.44% achieved by EIC, 6.28% by PESC, and 5.97% by NEI. At the end (iteration 30), the best feasible value found by 2-OPT-C is 7.08% compared to 6.60% by EIC, 6.54% by PESC, and 6.29% by NEI.
>
> This improvement provided by 2-OPT-C is significant and practically meaningful. Improving annualized mean return by 0.48% on a financial portfolio would be viewed as a significant addition of value by a hedge fund, investment bank or financial advisor. On a base investment of 1M compounded over 10 years, this would result in an additional 87K in investment income. While a competing method (EIC or PESC) might find a portfolio with a return as good as the one found by OPT-C if allowed more evaluations of the objective and constraint, this might not be possible when interacting with a client asking for a custom portfolio design as evaluating the objective and constraint 30 times already consumes roughly 6 hours of computation.
>
> |Portfolio Optimization (annualized return)|EIC|PESC|NEI|2-OPT-C|
> |:---|:----:|:---:|:---:|---:|
> |Iteration 5|5.95 %|5.79 %|5.62 %|6.02 %|
> |Iteration 15|6.44 %|6.28 %|5.97 %|6.83 %|
> |Iteration 30|6.60 %|6.54 %|6.29 %|7.08 %|
>
> ***Robot Pushing Problem:***
>
> We consider a robot pushing problem based on [Wang and Jegelka, 2017], where the robot pushes an object from the origin, i.e., $L_{\text{object}} = (0,0)$, to an unknown target location $L_{\text{target}} \in [-5,5]^2$. The parameters to be optimized are the location of the robot, i.e., $L_{\text{robot}} = (x_{\text{robot}}, y_{\text{robot}})$ and the duration of the push $t \in [1, 30]$. Therefore, the decision variables to optimize are $\set{x_{\text{robot}}, y_{\text{robot}}, t}$. The objective is to minimize the distance between the location of the object after being pushed and the target location, namely the $L_2$ norm of  $L_{\text{object after push}} -  L_{\text{target}}$. The cost function is the energy used by the robot for pushing the object which is $|| L_{\text{object after push}}|| + \epsilon$, where $\epsilon$ is some noise.
>
> We ran 50 replications for this experiment. For each experiment, we uniformly draw a target location. In other words, we have 50 different target locations uniformly distributed across the replications performed. In each replication, we start with three initial points and run 50 iterations. Here we benchmarked four algorithms NEI, EIC, PESC, and 2-OPT-C. (We also have experiment results for NEI on the portfolio optimization problem and will add that to the final version of the paper. 2-OPT-C outperformed it.) Here, we report the median of the distance between the object after being pushed and the target location of 50 replications.
>
> In the beginning, the median distances for the four algorithms are in the range of 2.5 to 2.6. After 25 iterations, the median distances for NEI and EIC are roughly 1.2 and 0.6, while PESC and 2-OPT-C achieve a median distance of 0.3. After we exhaust the evaluation budget, NEI has a median distance of 0.86 and the median distance for the EIC is 0.35. In comparison, the PESC and 2-OPT-C are able to achieve a smaller median distance of 0.12.
>
> |Robot Pushing|EIC|PESC|NEI|2-OPT-C|
> |:---|:----:|:---:|:---:|---:|
> |Iteration 25|0.6|0.3|1.2|0.3|
> |Iteration 50|0.35|0.12|0.86|0.12|
>
> References:
>
> [Ariafar et al., 2019] Setareh Ariafar, Jaume Coll-Font, Dana Brooks, and Jennifer Dy. Admmbo: Bayesian optimization with unknown constraints using admm. Journal of Machine Learning Research, 20(123):1–26, 2019.
>
> [Letham et al., 2019] Benjamin Letham, Brian Karrer, Guilherme Ottoni, Eytan Bakshy, et al. Constrained bayesian optimization with noisy experiments. Bayesian Analysis, 14(2):495–519, 2019.
>
> [Picheny et al., 2016] Victor Picheny, Robert B Gramacy, Stefan Wild, and Sebastien Le Digabel. Bayesian op- timization under mixed constraints with a slack-variable augmented lagrangian. In D. Lee, M. Sugiyama, U. Luxburg, I. Guyon, and R. Garnett, editors, Advances in Neural Information Processing Systems, volume 29, pages 1435–1443. Curran Associates, Inc., 2016.
>
> [Wang and Jegelka, 2017] Zi Wang and Stefanie Jegelka. Max-value entropy search for efficient Bayesian optimization. In International Conference on Machine Learning, pp. 3627–3635, 2017

---

### Official Review · Reviewer_8KpE · 2021-07-18

**Rating:** 5
**Confidence:** 4

**Summary:**

This paper proposes a 2-step lookahead variant of constrained EI [4,5] and builds on the unconstrained 2-step method of [17]. The introduction of constraints causes discontinuity to the acquisition function surface, which requires a novel treatment based on likelihood ratio and a main contribution of this work.

On the flip side, the empirical evaluation can be considerably improved:
(1) Given that there are already several constrained BO algorithms proposed in the literature, one would naturally expect real-world experiments to be appropriately motivated and performed instead of simply numerical experiments; this will allow us to know whether the proposed constrained BO method can perform well on real-world optimization tasks.
(2) Related to point 1, how can we tell whether the performance difference is significant in practice?
(3) In lines 166-182, the authors have described the importance of being non-myopic in CBO. How can we know whether the better empirical performance of their proposed approach is due to exactly this reason?

The argument on the importance of being non-myopic in constrained BO is not particularly motivating and requires further clarification:
(1) In terms of empirical validation, see point 3 above.
(2) Isn't it possible to modify PF in EIC to explore infeasible points more aggressively?
(3) On the other hand, supposing that the cost of sampling infeasible points is much higher (e.g., violation of safety or penalty for recommending infeasible points (line 295)), does this make 2-OPT-C less desirable than EIC?

Can the authors discuss how they would scale up to multi-step lookahead instead of simply two?

My other comments and concerns are detailed below.

**Limitations And Societal Impact:**

See the summary for the limitations of this work.

**Main Review:**

POST-REBUTTAL FEEDBACK

I like to thank the authors for their clarifications, some of which should be incorporated into the revised paper to ease readability. I have the follow-up questions on their response, but I am not expecting the authors to answer them with the limited time remaining. The authors can consider them for their revised paper.

A1 : For the portfolio optimization problem, what exactly is the constraint that you have imposed on the portfolio's risk (i.e., standard deviation of the return)? How would the results vary by relaxing/tightening this constraint, depending on the risk attitude of the investor?
Also, I understand that there are a number of works that have considered this problem in their experiments, such as the following work considering value at risk and variants in Bayesian optimization and the references therein:

Bayesian optimization of risk measures. NeurIPS 2020.

It is not clear why the proposed approach would be qualitatively and quantitatively better than the existing approaches like the above in this problem involving risk. The authors can consider expounding on this, which will help in the motivation of their work.

A3 : The authors say that "While one could design a heuristic acquisition function that puts extra weight on exploring infeasible points, this has not been previously proposed in the literature." Would this be a trivial modification? If so, the authors are encouraged to perform empirical comparison in this regard.

A4 : The authors say that "Our paper focuses on problems where the costs of sampling do not vary significantly across the domain." Can the authors provide the motivation of their problem in this regard? For example, why does the portfolio optimization problem adhere to this assumption?

A6 : I would encourage the authors to add this assumption to their problem setting so that a reader knows when to prefer their proposed approach.

A12: But delta remains strictly positive based on your proposed sequence. Wouldn't it miss sampling the global optimum then?

PRIOR FEEDBACK

Lines 50-52: The authors say that "This approach [1] requires an extremely large amount of computation to approximate the multi-step lookahead policy well, especially in problems with more than a few dimensions. This limits its applicability." This argument appears problematic in three ways: (1) What specific real-world applications have the authors motivated and performed in their experiments requiring higher computational efficiency, considering that the function evaluations in BO problems are typically costly? (2) Would the approach of [1] still be expensive if it is reduced to the context of 2-step lookahead? (3) Does the approach of [1] not trade off well between optimization performance and computational efficiency?

How can the proposed method be extended to one that is a truly batch mode where the second stage involves a batch of also q points to be evaluated?

It would be preferable that the authors specify exactly the form of the posterior distributions for E_0 and E_1 in the equation after line 152 to ease understanding. For example, the E_0's on the LHS and RHS do not appear to correspond to the same posterior distribution, albeit conditioned on D.

Page 4: Can the authors explain why do the formulations for constructing 2-OPT-C(X_1) not consider (...)^+ like that of EI?

Line 157: By setting delta > 0 in the theoretical analysis, isn't it possible that the proposed acquisition function would miss sampling the global optimum?

Can the authors provide a time complexity analysis for optimizing 2-POT-C?

Section 6: There is a lack of description of the three benchmark problems: Do the unconstrained conventional and non-myopic BO algorithms perform poorly in them?

Section 6.1: Can the authors give the mathematical expression of the median utility gap?

Lines 293-296: Do the authors mean that they follow the setup of [1] by penalizing infeasibility of recommended point?


The following references on conventional and batch BO with multi-step lookahead are missing:

Sequential Bayesian optimisation for spatial-temporal monitoring. UAI 2014.

Gaussian process planning with Lipschitz continuous reward functions: Towards unifying Bayesian optimization, active learning, and beyond. AAAI 2016.

Nonmyopic Gaussian process optimization with macro-actions. AISTATS 2020.



Minor issues

Line 75: difficulties requiring created by

Line 147: italicize D

**Time Spent Reviewing:**

~6

---

> ### Author Response · Authors · 2021-08-10
> **Author response to reviewer 8KpE featuring real world experiment**
>
> We thank the reviewer for the review and for taking the time to provide detailed feedback. We responded to the most significant concern, about a lack of real-world problems in our evaluation, by adding a new real-world experiment using financial portfolio optimization. We also feel that we have fully addressed concerns raised about motivation through simple clarifications that we will include in the final version of the manuscript. Finally, we responded to all requests for clarification. We feel that we have addressed all of the concerns raised in the review and hope that the reviewer will consider raising the rating for our paper. Due to the character limit, we post answers to Q9 - Q17 in Part 2
>
> $\bf{Q1:}$ Empirical evaluation can be considerably improved ... how can we tell whether the performance difference is significant in practice?
>
> $\bf{A1:}$
> Since reading these reviews, we have completed initial experiments studying a real-world portfolio optimization problem based on Boyd (2017), as studied in a BO context by Astudillo & Frazier (2020). In this problem, we optimize an algorithmic trading strategy to maximize a portfolio’s mean annualized return while constraining the portfolio’s risk, as measured by the standard deviation of the return. The portfolio simulation uses CVXPortfolio and real-world market data, as described in Boyd (2017). A single function evaluation using CVXPortfolio takes between 10 and 15 minutes. 50 replications have been completed of both 2-OPT-C, EIC and PESC, using 30 iterations and 3 initial points. At least one of the initial points is feasible.
> We find that 2-OPT-C provides a significant improvement. At iteration 0, the mean annualized return of the best feasible trading strategy for each of the three methods is near 5.1%. At an early stage (iteration 5), the three methods have similar performance with 2-OPT-C at  6.02%, EIC at 5.95%, PESC at 5.79%. Afterwards, 2-OPT-C begins to outperform the other two methods by a larger margin as the optimization proceeds. At iteration 15, 2-OPT-C achieves a value of 6.83% compared to the value 6.44% achieved by EIC and 6.28% by PESC. At the end (iteration 30), the best feasible value found by 2-OPT-C is 7.08% compared to 6.60% by EIC and 6.54% by PESC.
> This improvement provided by 2-OPT-C is significant and practically meaningful. Improving annualized mean return by 0.48% on a financial portfolio would be viewed as a significant addition of value by a hedge fund, investment bank or financial advisor. On a base investment of \\$1M compounded over 10 years, this would result in an additional $87K in investment income. While a competing method (EIC or PESC) might find a portfolio with a return as good as the one found by OPT-C if allowed more evaluations of the objective and constraint, this might not be possible when interacting with a client asking for a custom portfolio design as evaluating the objective and constraint 30 times already consumes roughly 6 hours of computation.
> We will include this experiment in the final version of the paper.
> We also plan to run experiments on at least one additional real-world problem, such as the reacting flow problem from [1], the hyperparameter tuning of MLP on MNIST data, or the robot pushing problem described in Z. Wang and S. Jegelka (2017).
>
> $\bf{Q2:}$
> In lines 166-182... exactly this reason?
>
> $\bf{A2:}$
> Together, the following four facts are all consistent with the empirical performance improvement being due to the non-myopic nature of our acquisition function: (1) the empirical performance improvement is substantial, (2) the difference between our acquisition function and EIC is that we look further ahead, (3) the explanation in lines 166-182 suggest that being non-myopic is likely to improve performance, (4) non-myopic seems to improve performance in unconstrained problems. Moreover, no other hypotheses present themselves explaining the performance improvement.
>
> We will additionally report how often 2-OPT-C and EIC evaluate a point whose probability of infeasibility is close to ½. The explanation in lines 166-182 predicts that 2-OPT-C should measure such points more often than EIC.
>
> $\bf{Q3:}$ Isn't it possible to modify PF in EIC to explore infeasible points more aggressively?
>
> $\bf{A3:}$
> While one could design a heuristic acquisition function that puts extra weight on exploring infeasible points, this has not been previously proposed in the literature. Moreover, many heuristics fail to be robust to corner cases and require tunable parameters that make the method difficult to use. Indeed, it is not always better to explore infeasible points --- it is only better in situations selected by a good multi-step lookahead strategy. For this reason, our approach is likely to be easier to use and to perform better than such a heuristic.
>
> $\bf{Q4:}$ Supposing that the cost of sampling infeasible points is much higher (e.g., violation of safety or penalty for recommending infeasible points (line 295)), does this make 2-OPT-C less desirable than EIC?
>
> $\bf{A4:}$ Our paper focuses on problems where the costs of sampling do not vary significantly across the domain. If sampling costs are heterogeneous, then instead of using 2-OPT-C or EIC one should use an acquisition function designed for this setting. One simple way to extend an acquisition function to the heterogeneous cost setting is to divide the acquisition function value by the cost of the samples. It would be interesting in future work to determine whether this idea or other ideas in our paper can be used to significantly improve the state of the art for constrained BO with heterogeneous costs.
>
> $\bf{Q5:}$ Can the authors discuss how they would scale up to multi-step lookahead instead of simply two?
>
> $\bf{A5:}$
> Scaling to more than 2 steps ahead is likely to significantly slow performance, as it would not obviously permit the special-purpose methods we have developed for 2-step lookahead. We argue that 2-step lookahead has an important role because it offers a significant improvement in query efficiency over being myopic while consuming significantly less computation than general-purpose multi-step lookahead methods.
>
> $\bf{Q6:}$ Lines 50-52: ... considering that the function evaluations in BO problems are typically costly?
>
> $\bf{A6:}$
> There are medium-cost problems where the function evaluations *are* costly, but fast enough that one cannot afford to spend a long time optimizing the acquisition function.
>
> For example, consider a problem where each evaluation takes 1 hour. Such problems are widespread in constrained black-box optimization. One specific example is optimizing the shape of an aircraft’s wing to maximize lift subject to a constraint on drag, using a computational fluid dynamics simulation of airflow over the wing that solves a large-scale partial differential equation. For details, see section IV.C of Thomison & Allaire (2017).
>
> Thomison, William D., and Douglas L. Allaire. "A model reification approach to fusing information from multifidelity information sources." 19th AIAA non-deterministic approaches conference. 2017.
>
> Suppose that a myopic method like EIC can achieve a good global solution in 60 iterations requiring 1 minute of overhead per acquisition function optimization, requiring 60*(60+1) minutes = 61 hours in total. Suppose that a non-myopic method implemented to optimize the non-myopic acquisition function extremely well can achieve the same quality in 30 iterations. Then, if the non-myopic method requires 1 hour of overhead per iteration, the problem can be solved in 30*(60+60) = 60 hours, which provides only marginal savings over a myopic method. If this same method were to require 20 minutes of overhead per iteration, then the problem is solved in 30*(60+20) = 40 hours, reducing compute time by 35%.
>
> $\bf{Q7:}$ Would the approach of [1] still be expensive if it is reduced to the context of 2-step lookahead?
>
> $\bf{A7:}$
> Yes, to optimize the 2-step lookahead acquisition function as well as we do with 2-OPT-C using the method in [1] would be quite expensive.
>
> The method of [1] has several key hyperparameters that govern how well it optimizes the acquisition function looking ahead a given number of steps: how many Monte Carlo samples to use in rollout, and when optimizing the acquisition function how many starts to use in multi-start gradient ascent and how many iterations of the optimization solver to use in each start. When these are set to large values, the quality of the acquisition function optimization improves but the computation time increases.
>
> When these hyperparameters are set to default values, the method of [1] takes approximately 20 minutes to optimize the constrained 2-step lookahead acquisition function, but does not do so very accurately. As a result, its query efficiency degrades.
>
> To reach the same query efficiency as our method, we one would need to set much larger hyperparameters in the method of [1]. While the code for [1] is unavailable in the constrained setting, we hypothesize that the code would take several hours to run with these hyperparameters.
>
> $\bf{Q8:}$ Does the approach of [1] not trade off well between optimization performance and computational efficiency?
>
> $\bf{A8:}$
> That is correct. Because it uses derivative-free optimization of a Monte Carlo estimate of the acquisition function, it is unable to optimize the acquisition function as well as our gradient-based method with a comparable amount of computation.  Please see lines 5-7 and 212-214.

---

> > ### Author Response · Authors · 2021-08-11
> > **Part 2**
> >
> > $\bf{Q9:}$
> > How can the proposed method be extended to one that is a truly batch mode where the second stage involves a batch of also q points to be evaluated?
> >
> > $\bf{A9:}$
> > Our method can extend to the settings where the acquisition function assumes that the second stage will use a batch of q points. This might improve the query efficiency of our method if done in a computationally accurate way. However, the value function resulting from the second stage’s batch of points (which is the constrained expected improvement) does not have a closed-form expression. Thus, we would require simulation to estimate the value in the second stage, similar to the method used in [1]. This would significantly increase the computation time required to optimize the acquisition function well. As in [1], if users were unwilling to incur this large computational time, it would require us to optimize the acquisition function less accurately, which would harm query efficiency. Thus, we view our current method as achieving an excellent balance between query efficiency and computation time.
> >
> > $\bf{Q10:}$
> > It would be preferable that the authors specify exactly the form of the posterior distributions for $E_0$ and $E_1$ in the equation after line 152 to ease understanding. For example, the $E_0$'s on the LHS and RHS do not appear to correspond to the same posterior distribution, albeit conditioned on $D$.
> >
> > $\bf{A10:}$
> > $E_0$ is explained in lines 139-40. It is the expectation taken with respect to the posterior given $D$.
> > $E_1$ is explained in lines 146-7. It is the expectation taken with respect to the posterior given $D$, $X_1$, the evaluations at $X_1$ ($Y_f$ and $Y_g$). We will clarify the sentence that explains $E_1$ to make explicit that $E_1$ is the posterior conditions not just on $X_1$ but also the evaluations $Y_f$ and $Y_g$.
> > In the final version of the paper we can write the posterior mean and kernel for these posterior distributions explicitly to support understanding.
> > The $E_0$’s on the LHS and RHS actually do correspond to the same posterior distribution. Both are conditioned on $D$.
> > To help support understanding of the equation after line 152, we include the following steps showing that the LHS is equal to the RHS.
> >
> > $$
> > E_0[\max_{x_2}(f_0^* - f_2^*)]
> > = E_0[ E_1[\max_{x_2} (f_0^* - f_1^* + f_1^* - f_2^*) | f(X_1), g(X_1)]]
> > = E_0[ f_0^* - f_1^* + \max_{x_2} E_1 (f_1^* - f_2^*)]$$
> >
> > The second line uses the law of total expectation and also adds and subtracts $f_1^*$. The third line uses that $f_0^*$ and $f_1^*$ are known constants under the posterior distribution over which $E_1$ takes the expectation.
> >
> > $\bf{Q11:}$ Can the authors explain why do the formulations for constructing 2-OPT-C(X_1) not consider (...)^+ like that of EI?
> >
> > $\bf{A11:}$ In line 149, $f_1^*$ and $f_2^*$ are the best evaluated feasible point by the end of the first and second stage, respectively. Thus, $$f_1^* = \min(f_0^*, \min f(X_1))$$
> > Therefore, $$f_0^* - f_1^* \geq 0$$ and we don’t need (...)^+.
> >
> > $\bf{Q12:}$ By setting delta > 0 in the theoretical analysis, isn't it possible that the proposed acquisition function would miss sampling the global optimum?
> >
> > $\bf{A12:}$
> > To avoid this risk, one can reduce delta over time, setting it a sequence of strictly positive values that decrease to 0.  For example, one may set delta in iteration n to 1/n. We will discuss this briefly in the paper.
> >
> > $\bf{Q13:}$
> > Can the authors provide a time complexity analysis for optimizing 2-OPT-C?
> >
> > $\bf{A13:}$
> > We currently include a detailed empirical discussion of the computation time required to optimize 2-OPT-C in the supplement. We will add to this to include a theoretical time-complexity analysis in the supplement in the final version of the paper. This will simply discuss the computation time required for each sample gradient obtained using likelihood ratio estimator and then reference convergence rates for stochastic gradient descent.
> >
> > $\bf{Q14:}$ There is a lack of description of the three benchmark problems: Do the unconstrained conventional and non-myopic BO algorithms perform poorly in them?
> >
> > $\bf{A14:}$
> > Due to the page limit, we put the experimental setup and details in the Appendix. The description of the three benchmark problems is provided in the Section A.2 of the appendix. For these three problems, the global optimum without the constraints is different from the global optimum under the constraints. Therefore, unconstrained BO algorithms will struggle to find the global optimum under the constraints. However, we are happy to run unconstrained BO methods on those problems in the final version of the paper. 2-OPT-C and all benchmarks currently included in the paper are likely to significantly outperform these new benchmarks.
> >
> > $\bf{Q15:}$
> > Can the authors give the mathematical expression of the median utility gap?
> >
> > $\bf{A15:}$ The utility gap is defined in the Section A.1 of the Appendix.
> >
> > $\bf{Q16:}$ Do the authors mean that they follow the setup of [1] by penalizing infeasibility of recommended point?
> >
> > $\bf{A16:}$ Yes.
> >
> > $\bf{Q17:}$ The following references....
> >
> > $\bf{A17:}$
> > Thank you.  We will add these references where we discuss other past work on unconstrained BO with multi-step lookahead.

---

> ### Author Response · Authors · 2021-09-01
> **Author response to POST-REBUTTAL FEEDBACK with new numerical results Part 1**
>
>
> Thank you for your questions.  We respond to them fully below. In addition, we include details of additional experiments performed during the response period that add new strong benchmarks from the literature and an additional problem motivated by a robotics application. 2-OPT-C performs quite well in these new experiments. Due to the character limit, we post the new numerical results in responses to POST-REBUTTAL FEEDBACK Part 2.
>
> $\bf{Q1:}$ For the portfolio optimization problem, what exactly is the constraint that you have imposed on the portfolio's risk (i.e., standard deviation of the return)? How would the results vary by relaxing/tightening this constraint, depending on the risk attitude of the investor?
>
> $\bf{A1:}$Yes, the constraint is on the yearly standard deviation of the return.
>
> We believe that our method provides the most value when at least one constraint is tight, as we explain in our paper in “The Importance of Being Non-Myopic in CBO”. In portfolio optimization problems, risk constraints are almost always tight, i.e. the risk is equal to the constraint under the optimal portfolio or trading strategy.  In other words, investors are rarely so immune to risk that they ignore it when planning their portfolios. Indeed, when trading equities, if one is allowed to short stocks, then not setting a risk constraint would lead to an infinitely large long position and an infinitely large short position with an infinite return --- this is impractical. Reflecting this reality, the constraint was set to be tight in our problem setting.
>
> If we were to relax or loosen the constraint, while still retaining the real-world property that the risk constraint is tight, we expect to see that 2-OPT-C will offer a significant benefit over competing methods.  This is because we have observed that 2-OPT-C does a better job than other methods of exploring the region where the constraint is tight and the objective is high.
>
> If we did not allow short selling and relaxed the problem further to an unrealistically large value of the risk constraint, so that it was no longer tight, then we expect to see that 2-OPT-C’s improvement over other methods would decline but not vanish. Indeed, in a robot pushing problem where the constraint is not tight at the optimum that we ran during the response period, 2-OPT-C continues to provide value over other methods. This may be because its non-myopic nature allows it to explore more effectively, as has been seen from non-myopic methods in other settings, and because its likelihood-ratio-based method for optimizing the acquisition function allows it to do so accurately. Please refer to the Additional Experiment section below for more details.
>
> $\bf{Q2:}$ "Also, I understand that there are a number of works that have considered this problem in their experiments, such as the following work considering value at risk and variants in Bayesian optimization and the references therein:
> Bayesian optimization of risk measures. NeurIPS 2020.
> It is not clear why the proposed approach would be qualitatively and quantitatively better than the existing approaches like the above in this problem involving risk. The authors can consider expounding on this, which will help in the motivation of their work."
>
> $\bf{A2:}$ The method in “Bayesian optimization of risk measures” assumes access to problem structure not present in most problems. In particular, it assumes that (1) the objective function is of the form rho(f(x,w)), where rho is either the value at risk or conditional value at risk, x is a decision variable, and w is a context variable; (2) f(x,w) can be evaluated for an individual x and w of the algorithm’s choosing; (3) w must be drawn from a finite or low-dimensional set to support computational tractability.
> Our method is better in that it is more general without the need for these special assumptions. While the specific portfolio problem we reported results on was one in which the method in “Bayesian optimization of risk measures” applies and can exploit problem structure to perform extremely well, small changes to the problem (e.g., increasing the dimensionality of w or changing to an expected shortfall risk measure) would make that method no longer apply while the characteristics that make our method perform well compared to benchmarks would not be significantly impacted.
>
> $\bf{Q3:}$  The authors say that "While one could design a heuristic acquisition function that puts extra weight on exploring infeasible points, this has not been previously proposed in the literature." Would this be a trivial modification? If so, the authors are encouraged to perform empirical comparison in this regard.
>
> $\bf{A3:}$ No, we don’t believe that designing a high-performance robust heuristic of this nature would be easy. While one of the reasons that our method provides significant improvement over the state of the art is because it is more willing to explore infeasible points, this must be done intelligently. For example, the amount of infeasibility that one should tolerate depends on the objective function value in this region and how it relates to the objective values at other points. One heuristic might be to find the point whose mean constraint value is equal to the constraint with the largest posterior mean, and to evaluate this point with some probability, but we expect this would perform quite badly in some settings when the optimum is not one where the constraint is tight. In contrast, our method’s performance does not degrade in such settings. Such heuristics also introduce tunable parameters that must be tuned, often in problem-specific ways. The method that we developed uses a principled approach to deliver state-of-the-art performance without the need for parameter tuning.
>
>
> $\bf{Q4:}$ The authors say that "Our paper focuses on problems where the costs of sampling do not vary significantly across the domain." Can the authors provide the motivation of their problem in this regard? For example, why does the portfolio optimization problem adhere to this assumption?
>
> $\bf{A4:}$There are many constrained optimization problems where the cost does not vary significantly across the domain. For example, the cost is homogeneous in the portfolio optimization problem because it is a fixed-length simulation in which the bulk of the work is in simulating the market. This does not depend on the trading strategy’s parameters. There is also some additional work in simulating the trading strategy itself but this work does not vary significantly with the trading strategy parameters.
>
> $\bf{Q5:}$ I would encourage the authors to add this assumption to their problem setting so that a reader knows when to prefer their proposed approach.
>
> $\bf{A5:}$ We will add it.
>
> $\bf{Q6:}$ But delta remains strictly positive based on your proposed sequence. Wouldn't it miss sampling the global optimum then?
>
> $\bf{A6:}$ First, we emphasize that delta can be set to an arbitrarily small value, e.g., $10^{-10}$. Then, when computing the acquisition function, while it is possible that the optimum in the second stage might lie within a ball of radius $10^{-10}$ near a previously sampled point, there will be a point that is slightly further away than $10^{-10}$ that is almost as good.  If one is worried that $10^{-10}$ may be too big, the value of delta can be set based on the kernel length-scale parameter to ensure that the loss in optimality is small. One can also set delta to a strictly positive sequence that decreases to 0 as the iteration number grows.
>
> Second, we emphasize that delta is only used to compute the acquisition function and does not imply an inability to sample close to previously sampled points.
>
> From a practical point of view, over a finite horizon setting, delta to $10^{-10}$ gives almost the same behavior as setting it to 0, and we recommend setting it to 0 in practice. In our theory, we choose delta > 0 for technical reasons to avoid a singularity induced by the inverse of standard deviation of posterior distribution at evaluated points. We currently believe (but have not shown) that our result also holds if delta = 0.

---

> > ### Author Response · Authors · 2021-09-01
> > **Author response to POST-REBUTTAL FEEDBACK with new numerical results Part 2**
> >
> > **New Experiments:**
> >
> > We conducted additional experiments during the rebuttal period to address the reviewers’ comments. We added four additional benchmark methods: ADMMBO [Ariafar et al., 2019], NEI [Letham et al., 2019], ALBO [Picheny etal., 2016] and random search (RS). We also added a more realistic problem, based on the robot pushing problem introduced in [Wang and Jegelka, 2017].  In addition, we added sample average approximation implemented using CMAES (SAA_CMAES) as a benchmark for problems P1 and P2. The result for P3 can be found in Figure 3 in the main paper. 2-OPT-C substantially outperforms benchmarks in all of these problems. Here we give details of these new experiments.
> >
> > ***New Benchmarks for Synthetic functions:***
> >
> > On synthetic problems, i.e., P1, P2, and P3 in the paper, we added the four additional methods mentioned above for comparison. We use the same experimental setup for these three problems as described in the Appendix. In detail, for each algorithm, we ran 150 replications on each problem with three initial points. We ran 40, 40, and 60 iterations for P1, P2, and P3, respectively, for each replication. The results are provided below.
> >
> > For P1, the (log10) median regret of each of the benchmarked algorithms starts between 0.3 and 0.4. After 20 iterations, RS, ADMMBO, ALBO, NEI, and SAA_CMAES achieve a median regret of 0, -0.18, -0.2, -2, and -2.3 respectively, while 2-OPT-C achieves a substantially smaller median regret of -3.6. After 40 iterations, the median regret of RS, ADMMBO, ALBO, NEI, and SAA_CMAES are -0.2, -0.5, -1.3, -2.2, and -2.9 respectively. The median regret of 2-OPT-C is substantially lower, at -4.8.
> >
> > | P1 (metric: log_10 median regret) | RS| ADMMBO|ALBO|NEI|SAA_CMAES|2-OPT-C|
> > |:---|:----:|:---:|:---:|:---:|:---:|:---:|
> > |Iteration 20| 0|-0.18|-0.2|-2.0|-2.3|-3.6|
> > |Iteration 40|-0.2|-0.5|-1.3|-2.2|-2.9|-4.8|
> >
> > For P2, the (log10) median regret of each of the benchmarked algorithms starts between -0.1 and -0.15. After 20 iterations, RS, ADMMBO, ALBO, NEI, and SAA_CMAES achieve a median utility of -0.6, -0.9, -1.25, -0.61, and -2.25 respectively, while 2-OPT-C achieves a much smaller median regret of -3. After 40 iterations, the median utility of RS, ADMMBO, ALBO, NEI, and SAA_CMAES are -0.75, -1.1, -1.8, -0.95, and -2.8 respectively. The median utility of 2-OPT-C is much smaller, at -3.2.
> >
> > | P2 (metric: log_10 median regret) | RS|ADMMBO|ALBO|NEI|SAA_CMAES|2-OPT-C|
> > |:---|:----:|:---:|:---:|:---:|:---:|:---:|
> > |Iteration 20| -0.6|-0.9|-1.25|-0.61|-2.25|-3.0|
> > |Iteration 40| -0.75|-1.1|-1.8|-0.95|-2.8|-3.2|
> >
> > For P3, the (log10) median regret of each of the benchmarked algorithms starts between 2.2 and 2.4. After 30 iterations, RS, NEI, and ALBO achieve similar median regret around 1.75, while ADMMBO is able to achieve a median regret of 1.48 and 2-OPT-C achieves an even smaller median regret of 1.35. After 60 iterations, the median regret of RS and ALBO is 1.67. The median regret of NEI is 1.51. Both ADMMBO and 2-OPT-C achieve the smallest median regret, 1.27.
> >
> > |P3 (metric: log_10 median regret)|RS|ADMMBO|ALBO|NEI|2-OPT-C|
> > |:---|:----:|:---:|:---:|:---:|:---:|
> > |Iteration 30| 1.75|1.48|1.75|1.75|1.35|
> > |Iteration 60| 1.67|1.27|1.67|1.51|1.27|
> >
> > In the experiments P1, P2, and P3, we notice that non-myopic policies (2-OPT-C and SAA_CMAES) generally perform better than myopic policies (ADMMBO, NEI, ALBO, EIC, and PESC). However, due to the discontinuity introduced by SAA, while SAA_CMAES outperforms the myopic methods in P2 and P3, it underperforms 2-OPT-C and also underperforms EIC in P1.
> >
> > ***Portfolio Optimization Problem.***
> >
> > This experiment studies a real-world portfolio optimization problem based on Boyd (2017), as studied in a BO context by Astudillo & Frazier (2020). In this problem, we optimize an algorithmic trading strategy to maximize a portfolio’s mean annualized return while constraining the portfolio’s risk, as measured by the standard deviation of the return. The portfolio simulation uses CVXPortfolio and real-world market data, as described in Boyd (2017). A single function evaluation using CVXPortfolio takes between 10 and 15 minutes. 50 replications have been completed for both 2-OPT-C, EIC and PESC, using 30 iterations and 3 initial points. At least one of the initial points is feasible.
> > We find that 2-OPT-C provides a significant improvement.
> >
> > At iteration 0, the mean annualized return of the best feasible trading strategy for each of the three methods is near 5.1%. At an early stage (iteration 5), 2-OPT-C and EIC have better performances with values at  6.02% and 5.95% respectively compared to PESC at 5.79% and NEI at 5.62%. Afterwards, 2-OPT-C begins to outperform the other three methods by a larger margin as the optimization proceeds. At iteration 15, 2-OPT-C achieves a value of 6.83% compared to the value of 6.44% achieved by EIC, 6.28% by PESC, and 5.97% by NEI. At the end (iteration 30), the best feasible value found by 2-OPT-C is 7.08% compared to 6.60% by EIC, 6.54% by PESC, and 6.29% by NEI.
> >
> > This improvement provided by 2-OPT-C is significant and practically meaningful. Improving annualized mean return by 0.48% on a financial portfolio would be viewed as a significant addition of value by a hedge fund, investment bank or financial advisor. On a base investment of 1M compounded over 10 years, this would result in an additional 87K in investment income. While a competing method (EIC or PESC) might find a portfolio with a return as good as the one found by OPT-C if allowed more evaluations of the objective and constraint, this might not be possible when interacting with a client asking for a custom portfolio design as evaluating the objective and constraint 30 times already consumes roughly 6 hours of computation.
> >
> > |Portfolio Optimization (annualized return)|EIC|PESC|NEI|2-OPT-C|
> > |:---|:----:|:---:|:---:|---:|
> > |Iteration 5|5.95 %|5.79 %|5.62 %|6.02 %|
> > |Iteration 15|6.44 %|6.28 %|5.97 %|6.83 %|
> > |Iteration 30|6.60 %|6.54 %|6.29 %|7.08 %|
> >
> > ***Robot Pushing Problem:***
> >
> > We consider a robot pushing problem based on [Wang and Jegelka, 2017], where the robot pushes an object from the origin, i.e., $L_{\text{object}} = (0,0)$, to an unknown target location $L_{\text{target}} \in [-5,5]^2$. The parameters to be optimized are the location of the robot, i.e., $L_{\text{robot}} = (x_{\text{robot}}, y_{\text{robot}})$ and the duration of the push $t \in [1, 30]$. Therefore, the decision variables to optimize are $\set{x_{\text{robot}}, y_{\text{robot}}, t}$. The objective is to minimize the distance between the location of the object after being pushed and the target location, namely the $L_2$ norm of  $L_{\text{object after push}} -  L_{\text{target}}$. The cost function is the energy used by the robot for pushing the object which is $|| L_{\text{object after push}}|| + \epsilon$, where $\epsilon$ is some noise.
> >
> > We ran 50 replications for this experiment. For each experiment, we uniformly draw a target location. In other words, we have 50 different target locations uniformly distributed across the replications performed. In each replication, we start with three initial points and run 50 iterations. Here we benchmarked four algorithms NEI, EIC, PESC, and 2-OPT-C. (We also have experiment results for NEI on the portfolio optimization problem and will add that to the final version of the paper. 2-OPT-C outperformed it.) Here, we report the median of the distance between the object after being pushed and the target location of 50 replications.
> >
> > In the beginning, the median distances for the four algorithms are in the range of 2.5 to 2.6. After 25 iterations, the median distances for NEI and EIC are roughly 1.2 and 0.6, while PESC and 2-OPT-C achieve a median distance of 0.3. After we exhaust the evaluation budget, NEI has a median distance of 0.86 and the median distance for the EIC is 0.35. In comparison, the PESC and 2-OPT-C are able to achieve a smaller median distance of 0.12.
> >
> > |Robot Pushing|EIC|PESC|NEI|2-OPT-C|
> > |:---|:----:|:---:|:---:|---:|
> > |Iteration 25|0.6|0.3|1.2|0.3|
> > |Iteration 50|0.35|0.12|0.86|0.12|
> >
> > References:
> >
> > [Ariafar et al., 2019] Setareh Ariafar, Jaume Coll-Font, Dana Brooks, and Jennifer Dy. Admmbo: Bayesian optimization with unknown constraints using admm. Journal of Machine Learning Research, 20(123):1–26, 2019.
> >
> > [Letham et al., 2019] Benjamin Letham, Brian Karrer, Guilherme Ottoni, Eytan Bakshy, et al. Constrained bayesian optimization with noisy experiments. Bayesian Analysis, 14(2):495–519, 2019.
> >
> > [Picheny et al., 2016] Victor Picheny, Robert B Gramacy, Stefan Wild, and Sebastien Le Digabel. Bayesian op- timization under mixed constraints with a slack-variable augmented lagrangian. In D. Lee, M. Sugiyama, U. Luxburg, I. Guyon, and R. Garnett, editors, Advances in Neural Information Processing Systems, volume 29, pages 1435–1443. Curran Associates, Inc., 2016.
> >
> > [Wang and Jegelka, 2017] Zi Wang and Stefanie Jegelka. Max-value entropy search for efficient Bayesian optimization. In International Conference on Machine Learning, pp. 3627–3635, 2017

---

### Author Response · Authors · 2021-09-02
**Author response to the area chairs and the reviewers**

We would like to start by thanking the reviewing team for the valuable feedback we have received. During the rebuttal period, we have fully addressed the questions raised by the reviewers and also added several new numerical experiments to alleviate the concerns the reviewers had about the experimental section. Since we believe these new experiments would make our submission much stronger and are of interest to the whole reviewing team, we describe them in detail below along with the results obtained. We sincerely hope the individual responses we provided during the rebuttal period and these new experimental results help improve the reviewers’ opinion of our work.

**New Experiments:**

We conducted additional experiments during the rebuttal period to address the reviewers’ comments. We added four additional benchmark methods: ADMMBO [Ariafar et al., 2019], NEI [Letham et al., 2019], ALBO [Picheny etal., 2016] and random search (RS). We also added a more realistic problem, based on the robot pushing problem introduced in [Wang and Jegelka, 2017].  In addition, we added sample average approximation implemented using CMAES (SAA_CMAES) as a benchmark for problems P1 and P2. The result for P3 can be found in Figure 3 in the main paper. 2-OPT-C substantially outperforms benchmarks in all of these problems. Here we give details of these new experiments.

***New Benchmarks for Synthetic functions:***

On synthetic problems, i.e., P1, P2, and P3 in the paper, we added the four additional methods mentioned above for comparison. We use the same experimental setup for these three problems as described in the Appendix. In detail, for each algorithm, we ran 150 replications on each problem with three initial points. We ran 40, 40, and 60 iterations for P1, P2, and P3, respectively, for each replication. The results are provided below.

For P1, the (log10) median regret of each of the benchmarked algorithms starts between 0.3 and 0.4. After 20 iterations, RS, ADMMBO, ALBO, NEI, and SAA_CMAES achieve a median regret of 0, -0.18, -0.2, -2, and -2.3 respectively, while 2-OPT-C achieves a substantially smaller median regret of -3.6. After 40 iterations, the median regret of RS, ADMMBO, ALBO, NEI, and SAA_CMAES are -0.2, -0.5, -1.3, -2.2, and -2.9 respectively. The median regret of 2-OPT-C is substantially lower, at -4.8.

| P1 (metric: log_10 median regret) | RS| ADMMBO|ALBO|NEI|SAA_CMAES|2-OPT-C|
|:---|:----:|:---:|:---:|:---:|:---:|:---:|
|Iteration 20| 0|-0.18|-0.2|-2.0|-2.3|-3.6|
|Iteration 40|-0.2|-0.5|-1.3|-2.2|-2.9|-4.8|

For P2, the (log10) median regret of each of the benchmarked algorithms starts between -0.1 and -0.15. After 20 iterations, RS, ADMMBO, ALBO, NEI, and SAA_CMAES achieve a median utility of -0.6, -0.9, -1.25, -0.61, and -2.25 respectively, while 2-OPT-C achieves a much smaller median regret of -3. After 40 iterations, the median utility of RS, ADMMBO, ALBO, NEI, and SAA_CMAES are -0.75, -1.1, -1.8, -0.95, and -2.8 respectively. The median utility of 2-OPT-C is much smaller, at -3.2.

| P2 (metric: log_10 median regret) | RS|ADMMBO|ALBO|NEI|SAA_CMAES|2-OPT-C|
|:---|:----:|:---:|:---:|:---:|:---:|:---:|
|Iteration 20| -0.6|-0.9|-1.25|-0.61|-2.25|-3.0|
|Iteration 40| -0.75|-1.1|-1.8|-0.95|-2.8|-3.2|

For P3, the (log10) median regret of each of the benchmarked algorithms starts between 2.2 and 2.4. After 30 iterations, RS, NEI, and ALBO achieve similar median regret around 1.75, while ADMMBO is able to achieve a median regret of 1.48 and 2-OPT-C achieves an even smaller median regret of 1.35. After 60 iterations, the median regret of RS and ALBO is 1.67. The median regret of NEI is 1.51. Both ADMMBO and 2-OPT-C achieve the smallest median regret, 1.27.

|P3 (metric: log_10 median regret)|RS|ADMMBO|ALBO|NEI|2-OPT-C|
|:---|:----:|:---:|:---:|:---:|:---:|
|Iteration 30| 1.75|1.48|1.75|1.75|1.35|
|Iteration 60| 1.67|1.27|1.67|1.51|1.27|

In the experiments P1, P2, and P3, we notice that non-myopic policies (2-OPT-C and SAA_CMAES) generally perform better than myopic policies (ADMMBO, NEI, ALBO, EIC, and PESC). However, due to the discontinuity introduced by SAA, while SAA_CMAES outperforms the myopic methods in P2 and P3, it underperforms 2-OPT-C and also underperforms EIC in P1.

***Portfolio Optimization Problem.***

This experiment studies a real-world portfolio optimization problem based on Boyd (2017), as studied in a BO context by Astudillo & Frazier (2020). In this problem, we optimize an algorithmic trading strategy to maximize a portfolio’s mean annualized return while constraining the portfolio’s risk, as measured by the standard deviation of the return. The portfolio simulation uses CVXPortfolio and real-world market data, as described in Boyd (2017). A single function evaluation using CVXPortfolio takes between 10 and 15 minutes. 50 replications have been completed for both 2-OPT-C, EIC and PESC, using 30 iterations and 3 initial points. At least one of the initial points is feasible.
We find that 2-OPT-C provides a significant improvement.

At iteration 0, the mean annualized return of the best feasible trading strategy for each of the three methods is near 5.1%. At an early stage (iteration 5), 2-OPT-C and EIC have better performances with values at  6.02% and 5.95% respectively compared to PESC at 5.79% and NEI at 5.62%. Afterwards, 2-OPT-C begins to outperform the other three methods by a larger margin as the optimization proceeds. At iteration 15, 2-OPT-C achieves a value of 6.83% compared to the value of 6.44% achieved by EIC, 6.28% by PESC, and 5.97% by NEI. At the end (iteration 30), the best feasible value found by 2-OPT-C is 7.08% compared to 6.60% by EIC, 6.54% by PESC, and 6.29% by NEI.

This improvement provided by 2-OPT-C is significant and practically meaningful. Improving annualized mean return by 0.48% on a financial portfolio would be viewed as a significant addition of value by a hedge fund, investment bank or financial advisor. On a base investment of 1M compounded over 10 years, this would result in an additional 87K in investment income. While a competing method (EIC or PESC) might find a portfolio with a return as good as the one found by OPT-C if allowed more evaluations of the objective and constraint, this might not be possible when interacting with a client asking for a custom portfolio design as evaluating the objective and constraint 30 times already consumes roughly 6 hours of computation.

|Portfolio Optimization (annualized return)|EIC|PESC|NEI|2-OPT-C|
|:---|:----:|:---:|:---:|---:|
|Iteration 5|5.95 %|5.79 %|5.62 %|6.02 %|
|Iteration 15|6.44 %|6.28 %|5.97 %|6.83 %|
|Iteration 30|6.60 %|6.54 %|6.29 %|7.08 %|

***Robot Pushing Problem:***

We consider a robot pushing problem based on [Wang and Jegelka, 2017], where the robot pushes an object from the origin, i.e., $L_{\text{object}} = (0,0)$, to an unknown target location $L_{\text{target}} \in [-5,5]^2$. The parameters to be optimized are the location of the robot, i.e., $L_{\text{robot}} = (x_{\text{robot}}, y_{\text{robot}})$ and the duration of the push $t \in [1, 30]$. Therefore, the decision variables to optimize are $\set{x_{\text{robot}}, y_{\text{robot}}, t}$. The objective is to minimize the distance between the location of the object after being pushed and the target location, namely the $L_2$ norm of  $L_{\text{object after push}} -  L_{\text{target}}$. The cost function is the energy used by the robot for pushing the object which is $|| L_{\text{object after push}}|| + \epsilon$, where $\epsilon$ is some noise.

We ran 50 replications for this experiment. For each experiment, we uniformly draw a target location. In other words, we have 50 different target locations uniformly distributed across the replications performed. In each replication, we start with three initial points and run 50 iterations. Here we benchmarked four algorithms NEI, EIC, PESC, and 2-OPT-C. (We also have experiment results for NEI on the portfolio optimization problem and will add that to the final version of the paper. 2-OPT-C outperformed it.) Here, we report the median of the distance between the object after being pushed and the target location of 50 replications.

In the beginning, the median distances for the four algorithms are in the range of 2.5 to 2.6. After 25 iterations, the median distances for NEI and EIC are roughly 1.2 and 0.6, while PESC and 2-OPT-C achieve a median distance of 0.3. After we exhaust the evaluation budget, NEI has a median distance of 0.86 and the median distance for the EIC is 0.35. In comparison, the PESC and 2-OPT-C are able to achieve a smaller median distance of 0.12.

|Robot Pushing|EIC|PESC|NEI|2-OPT-C|
|:---|:----:|:---:|:---:|---:|
|Iteration 25|0.6|0.3|1.2|0.3|
|Iteration 50|0.35|0.12|0.86|0.12|

References:

[Ariafar et al., 2019] Setareh Ariafar, Jaume Coll-Font, Dana Brooks, and Jennifer Dy. Admmbo: Bayesian optimization with unknown constraints using admm. Journal of Machine Learning Research, 20(123):1–26, 2019.

[Letham et al., 2019] Benjamin Letham, Brian Karrer, Guilherme Ottoni, Eytan Bakshy, et al. Constrained bayesian optimization with noisy experiments. Bayesian Analysis, 14(2):495–519, 2019.

[Picheny et al., 2016] Victor Picheny, Robert B Gramacy, Stefan Wild, and Sebastien Le Digabel. Bayesian op- timization under mixed constraints with a slack-variable augmented lagrangian. In D. Lee, M. Sugiyama, U. Luxburg, I. Guyon, and R. Garnett, editors, Advances in Neural Information Processing Systems, volume 29, pages 1435–1443. Curran Associates, Inc., 2016.

[Wang and Jegelka, 2017] Zi Wang and Stefanie Jegelka. Max-value entropy search for efficient Bayesian optimization. In International Conference on Machine Learning, pp. 3627–3635, 2017

---

### Decision · Program_Chairs · 2021-09-28

**Decision:**

Accept (Poster)

**Comment:**

This manuscript considers the development of nonmyopic Bayesian optimization policies for constrained optimization. Although nonmyopic policies have repeatedly shown success in unconstrained settings, there are several challenges to overcome in the constrained setting.

Over the course of the discussion phase, the reviewers came to the consensus that this paper was notable, offered a significant contribution, was of interest to the NeurIPS community, and could serve as a springboard for future work.

In preparing the final version of the manuscript, I strongly suggest that the authors take the reviewers' suggestions into account. Further, the reviewers agree that the the portfolio optimization example from the discussion should be incorporated into the paper, as it was both relevant and convincing.

**Consistency Experiment:**

NeurIPS has a long history of experimentation. In 2014, NeurIPS ran an experiment in which 10% of submissions were reviewed by two independent committees to quantify the randomness in the review process. This year, we repeated a variant of this experiment to see how the quality of the review process has changed over time.  This paper was part of the experiment and was therefore assigned to two committees (consisting of reviewers, an Area Chair, and a Senior Area Chair) that reached independent decisions.  If both committees made the same recommendation, this recommendation was followed. If a single committee recommended acceptance, the paper was accepted (with the exception of a few cases in which the other committee identified what we considered a fatal flaw, e.g., an error in a key result).

This copy’s committee reached the following decision: **Accept (Poster)**

The other committee assigned to the paper recommended **Reject**.  You can find the other set of reviews, along with any follow up discussion with the authors here:
https://openreview.net/forum?id=oVEGzC7ieOB